# Atmospheric $\Delta^{17}O(NO_3^-)$ reveals nocturnal chemistry dominates nitrate production in Beijing haze

Pengzhen He[1], Zhouqing Xie[1,2,3]*, Xiyuan Chi[1], Xiawei Yu[1], Shidong Fan[1], Hui Kang[1], Cheng Liu[1,2,3], Haicong Zhan[1]

[1]Anhui Province Key Laboratory of Polar Environment and Global Change, School of Earth and Space Sciences, University of Science and Technology of China, Hefei, Anhui 230026, China.

[2]Center for Excellence in Urban Atmospheric Environment, Institute of Urban Environment, Chinese Academy of Sciences, Xiamen, Fujian 361021, China.

[3]Key Lab of Environmental Optics and Technology, Anhui Institute of Optics and Fine Mechanics, Chinese Academy of Sciences, Hefei, Anhui 230031, China.

*Corresponding to*: Zhouqing Xie (zqxie@ustc.edu.cn)

**Abstract.** The rapid mass increase of atmospheric nitrate is a critical driving force for the occurrence of fine-particle pollution (referred to as haze hereafter) in Beijing. However, the exact mechanisms for this rapid increase of nitrate mass has been not well constrained from field observations. Here we present the first observations of the oxygen-17 excess of atmospheric nitrate ($\Delta^{17}O(NO_3^-)$) collected in Beijing haze to reveal the relative importance of different nitrate formation pathways, and we also present the simultaneously observed $\delta^{15}N(NO_3^-)$. During our sampling period, 12h-averaged mass concentrations of $PM_{2.5}$ varied from 16 to 323 μg m$^{-3}$ with a mean of (141±88 (1SD)) μg m$^{-3}$, with nitrate ranging from 0.3 to 106.7 μg m$^{-3}$. The observed $\Delta^{17}O(NO_3^-)$ ranged from 27.5 ‰ to 33.9 ‰ with a mean of (30.6±1.8) ‰ while $\delta^{15}N(NO_3^-)$ ranged from −2.5 ‰ to 19.2 ‰ with a mean of (7.4±6.8) ‰. $\Delta^{17}O(NO_3^-)$-constrained calculations suggest nocturnal pathways ($N_2O_5 + H_2O/Cl^-$ and $NO_3 + HC$) dominated nitrate production during polluted days ($PM_{2.5} \geq 75$ μg m$^{-3}$) with the mean possible fraction of 56 − 97 %. Our results illustrate the potentiality of $\Delta^{17}O$ in tracing nitrate formation pathways, future modelling work with the constraint of isotope data reported here may further improve our understanding of nitrogen cycle during haze.

## 1 Introduction

Severe and frequent haze pollution has become a crucial threat for the air quality in megacity Beijing and the North China Plain in recent years. The high concentrations of $PM_{2.5}$ (particulate matter with an aerodynamic diameter equal or less than 2.5 μm) during severe haze, of which the hourly average can reach 1000 μg m$^{-3}$ (Zheng et al., 2015a), is harmful

to the public health by contributing to cardiovascular morbidity and mortality (Cheng et al., 2013; Brook et al., 2010).

Nitrate is an important component of $PM_{2.5}$, accounting for 1–45 % of $PM_{2.5}$ mass in Beijing and North China Plain (Wen et al., 2015; Zheng et al., 2015a; Zheng et al., 2015b). The main formation pathways of atmospheric nitrate, defined herein as gas-phase $HNO_3$ plus particulate $NO_3^-$, in urban area are summarized in Fig. 1, which includes: (i) $NO_2$ oxidation by OH radical in the gas-phase, (ii) heterogeneous uptake of $NO_2$ on wet aerosols, (iii) $NO_3$ radical reacting with hydrocarbon (HC), and (iv) heterogeneous uptake of $N_2O_5$ on wet aerosols and chlorine-containing aerosols. Since OH radical is mainly present in the daytime while $NO_3$ radical and $N_2O_5$ are mainly present in the nocturnal atmosphere (Brown and Stutz, 2012), $NO_2$ + OH is usually referred as the daytime nitrate formation pathway while $N_2O_5$ + $H_2O/Cl^-$ and $NO_3$ + HC are referred as nocturnal formation pathways (Vicars et al., 2013; Sofen et al., 2014). During haze in Beijing, the mixing ratio of daytime OH is modelled to be low (Zheng et al., 2015b; Rao et al., 2016) while relatively high mixing ratio of nocturnal $N_2O_5$ is observed in several studies (Wang et al., 2017a; Li et al., 2018; Wang et al., 2017b), therefore, nocturnal pathways are suggested to be most responsible for the high concentrations of atmospheric nitrate during haze (Su et al., 2017; Pathak et al., 2009; Pathak et al., 2011). In addition, the high $PM_{2.5}$ concentration and relative humidity during haze in Beijing favors heterogeneous reactions, which renders $NO_2$ + $H_2O$ being a potentially significant pathway for nitrate production (Wang et al., 2017d; Tong et al., 2015; Zheng et al., 2015a).

Nitrogen isotopic composition of nitrate ($\delta^{15}N(NO_3^-)$, wherein $\delta^{15}N = (R_{sample}/R_{reference} - 1)$ with $R$ representing isotope ratios of $^{15}N/^{14}N$ in the sample and the reference atmospheric $N_2$) is useful in tracing source of its precursor $NO_X$ (Xiao et al., 2015; Beyn et al., 2014; Fang et al., 2011; Hastings et al., 2013). Anthropogenic sources of $NO_X$ such as coal combustion are generally enriched in $\delta^{15}N$ while natural $NO_X$ sources such as soil emissions or lighting typically have negative or zero $\delta^{15}N$ signature (Hoering, 1957; Yu and Elliott, 2017; Felix et al., 2012). Therefore highly positive values of observed $\delta^{15}N(NO_3^-)$ can be considered as an indicator of anthropogenic combustion (Elliott et al., 2009; Fang et al., 2011), although this judgment may be influenced by isotopic exchange between NO and $NO_2$ (Freyer et al., 1993; Walters et al., 2016), isotopic fractionations associated with nitrate formation pathways and isotopic effects occurring during transport, such as deposition of $NO_3^-$ and $HNO_3$ partitioning between gas and particle phase (Freyer, 1991; Geng et al., 2014). The oxygen-17 excess ($\Delta^{17}O$) of nitrate, defined as $\Delta^{17}O = \delta^{17}O - 0.52\delta^{18}O$, wherein $\delta^X O = (R_{sample}/R_{reference} - 1)$ with $R$ representing isotope ratios of $^X O/^{16}O$ in the sample and the reference Vienna Standard Mean Ocean Water and X = 17 or 18, is particularly useful in reflecting nitrate formation pathways (Michalski et al., 2003). Atmospheric nitrate from nocturnal reaction pathways has higher $\Delta^{17}O$ than that from daytime OH oxidation at given $\Delta^{17}O(NO_2)$ (Table 1). And once formed, atmospheric $\Delta^{17}O(NO_3^-)$ cannot be altered by mass-dependent processes such as deposition during transport (Brenninkmeijer et al., 2003). Previous studies have shown the utility of atmospheric $\Delta^{17}O(NO_3^-)$ in quantifying the relative importance of various nitrate formation pathways (Alexander et al., 2009; Michalski et al., 2003; Patris et al., 2007; Savarino et al., 2013; Vicars et al., 2013). For example, $\Delta^{17}O(NO_3^-)$-constrained box modeling work of Michalski et al. (2003) suggests that more than 90 % of

atmospheric nitrate is from nocturnal $N_2O_5 + H_2O$ pathway in winter La Jolla, California, which is reflected by the highest
$\Delta^{17}O(NO_3^-)$ values being observed in winter. In another study, Alexander et al. (2009) use observed $\Delta^{17}O(NO_3^-)$ to constrain
3D model and found that daytime $NO_2 + OH$ pathway dominates global tropospheric nitrate production with an annual mean
contribution of 76 %.
Until now, however, field observations of atmospheric $\Delta^{17}O(NO_3^-)$ have not been conducted in north China to constrain
the relative importance of different nitrate formation pathways during haze. In this work, we present the first observations of
atmospheric $\Delta^{17}O(NO_3^-)$ during Beijing haze from October 2014 to January 2015, and use this observation to examine the
importance of nocturnal formation pathways. We also present the signature of simultaneously observed $\delta^{15}N(NO_3^-)$.
**2 Materials and Methods**
**2.1 Sampling and atmospheric observations**
$PM_{2.5}$ filter samples were collected at a flow rate of 1.05 $m^3$ $min^{-1}$ by a high volume air sampler (model TH-1000C II,
Tianhong Instruments Co., Ltd, China). The filter is quartz microfiber filter (Whatman Inc., UK), pre-combusted at 450 °C
for 4 h before sampling. Our sampling period lasted from October 2014 to January 2015 with the collection interval being 12
h (08:00 – 20:00 LT or 20:00 – 08:00 LT) for each sample. Blank control samples were also collected. The blank was
sampled identically to the real sample except that the collection interval is 1 min. Due to that gaseous $HNO_3$ is likely to
adsorb onto particulate matter already trapped by the filter material (Vicars et al., 2013), the nitrate species collected is likely
to include both particulate nitrate and gaseous $HNO_3$, which is referred to as atmospheric nitrate in previous studies (Vicars
et al., 2013; Morin et al., 2009; Michalski et al., 2003) and in this study. The sampling site is at the campus of University of
the Chinese Academy of Sciences (40.41 ° N, 116.68 ° E, ~20 m high) in suburban Beijing, about 60 km northeast of
downtown (Fig. 2), which is a super site set by HOPE-J³A (Haze Observation Project Especially for Jing-Jin-Ji Area) with
various observations being reported (Zhang et al., 2017; Xu et al., 2016; Chen et al., 2015; Tong et al., 2015; He et al., 2018).
Hourly concentrations of surface $PM_{2.5}$, CO, $SO_2$, $NO_2$ and $O_3$ were observed at Huairou station (40.33 ° N, 116.63 ° E) by
Beijing Municipal Environmental Monitoring Center, about 10 km to our sampling site. Meteorological data including
relative humidity (RH) and air temperature ($T$) were measured by an automatic weather station (model MetPak, Gill
Instruments Limited, UK). Time used in the present study is local time (LT = UTC + 8).
**2.2 Measurements of ions and isotopic ratios**
Ion concentrations of $NO_3^-$ and $Cl^-$ were measured in Anhui Province Key Laboratory of Polar Environment and Global
Change in the University of Science and Technology of China. A detailed description of this method can be found in the
literature (Ye et al., 2015). Briefly, ions in the $PM_{2.5}$ filter sample were extracted with Millipore water ($\geq$ 18 M$\Omega$) and
insoluble substances in the extract were filtered. Then the ion concentrations were analyzed by an ion chromatograph system
(model Dionex ICS-2100, Thermo Fisher Scientific Inc., USA). The measured ion concentrations of blank samples were
subtracted when determining the ion concentrations of real samples. Typical analytical precision by our method is better than
10 % relative standard deviation (RSD) (Chen et al., 2016).

$\delta^{15}N(NO_3^-)$ and $\Delta^{17}O(NO_3^-)$ were measured with a bacterial denitrifier method (Kaiser et al., 2007) in IsoLab at the

University of Washington, USA. Briefly, ions in the filter sample were extracted with Millipore water ($\geq$ 18 M$\Omega$) and the
insoluble substances were filtered. $NO_3^-$ in each sample was converted to $N_2O$ by the denitrifying bacteria, Pseudomonas
aureofaciens. Then $N_2$ and $O_2$, which were decomposed from $N_2O$ in a gold tube at 800 ° C, were separated by a gas
chromatograph. The isotopic ratios of each gas were then measured by a Finnigan Delta-Plus Advantage isotope ratio mass
spectrometer. Masses of 28 and 29 from $N_2$ were measured to determine $\delta^{15}N$. Masses of 32, 33 and 34 from $O_2$ were
measured to determine $\delta^{17}O$ and $\delta^{18}O$ and $\Delta^{17}O$ was then calculated. We use international nitrate reference materials,
USGS34, USGS35 and IAEANO$_3$, for data calibration. The uncertainty (1$\sigma$) of $\delta^{15}N$ and $\Delta^{17}O$ measurements in our method
is 0.4 ‰ and 0.2 ‰, respectively, based on replicate analysis of the international reference materials. All the samples
including blank samples were measured in triplicate to quantify the uncertainty in each sample. The blank was subtracted for
each sample by using an isotopic mass balance on the basis of isotopic ratios and concentrations of the blank. To minimize
the blank effect, samples with blank concentrations being > 10 % of their concentrations were not analyzed for isotopic
ratios. This ruled out 3 of the total 34 samples, all of which are in non-polluted days (NPD, $PM_{2.5} < 75$ μg m$^{-3}$). Totally,
isotopic compositions of 7 samples in NPD and 24 samples in polluted days (PD, $PM_{2.5} \geq 75$ μg m$^{-3}$) are reported here.
**2.3 Estimate of different nitrate formation pathways based on $\Delta^{17}O(NO_3^-)$**

The observed $\Delta^{17}O(NO_3^-)$ is determined by the relative importance of different nitrate formation pathways and the

relative importance of $O_3$ oxidation in $NO_X$ cycling as shown in Eq. (1):
$\Delta^{17}O(NO_3^-) = \Delta^{17}O_{R6} \times f_{R6} + \Delta^{17}O_{R7} \times f_{R7} + \Delta^{17}O_{R8} \times f_{R8} + \Delta^{17}O_{R9} \times f_{R9} + \Delta^{17}O_{R10} \times f_{R10}$    (1)
Where $\Delta^{17}O_{R6}$, $\Delta^{17}O_{R7}$, $\Delta^{17}O_{R8}$, $\Delta^{17}O_{R9}$ and $\Delta^{17}O_{R10}$ is respectively $\Delta^{17}O(NO_3^-)$ resulting from $NO_2 + OH$, $NO_2 + H_2O$, $NO_3 +$
HC, $N_2O_5 + H_2O$ and $N_2O_5 + Cl^-$ pathway (Table 1). $f_{R6}$, $f_{R7}$, $f_{R8}$, $f_{R9}$ and $f_{R10}$ is respectively corresponding fractional
contribution of above pathways to nitrate production. By using the $\Delta^{17}O$ assumptions for different pathways in Table 1 and
the definition $f_{R6} + f_{R7} + f_{R8} + f_{R9} + f_{R10} = 1$, Eq. (1) is further expressed as:
$\Delta^{17}O(NO_3^-)/‰ = 25\alpha f_{R6} + 25\alpha f_{R7} + (25\alpha + 14) \times f_{R8} + (25\alpha + 7) \times f_{R9} + (25\alpha + 14) \times f_{R10} = 25\alpha + 14 \times$
$(f_{R8} + f_{R10}) + 7 f_{R9}$    (2)
Where $\alpha$ is the proportion of $O_3$ oxidation in $NO_2$ production rate, calculated by Eq. (3):
$\alpha = \dfrac{k_{R1}[NO][O_3]}{k_{R1}[NO][O_3]+k_{R2a}[NO][HO_2]+k_{R2b}[NO][RO_2]}$     (3)
In Eq. (3), $k_{R1}$, $k_{R2a}$ and $k_{R2b}$ is respectively the reaction rate constant listed in Table 2. To evaluate $\alpha$, we estimated $HO_2$
mixing ratios on the basis of empirical formulas between $HO_2$ and $O_3$ mixing ratios derived from observations in winter
(Kanaya et al., 2007), that's: $[HO_2]/(pmol\ mol^{-1}) = \exp(5.7747 \times 10^{-2} \times [O_3]/(nmol\ mol^{-1}) - 1.7227)$ during the day time and
$[HO_2]/(pmol\ mol^{-1}) = \exp(7.7234 \times 10^{-2} \times [O_3]/(nmol\ mol^{-1}) - 1.6363)$ at night. Then $RO_2$ mixing ratio was calculated as 70 %
of $HO_2$ mixing ratios based on previous studies (Liu et al., 2012; Elshorbany et al., 2012; Mihelcic et al., 2003). As NO
mixing ratio was not observed in our study, we estimated NO mixing ratios following the empirical formulas between $NO_X$
and CO mixing ratios derived from observations in winter Beijing (Lin et al., 2011), that's: $[NO]/(nmol\ mol^{-1}) =$
$([CO]/(nmol\ mol^{-1}) - 196)/27.3 - [NO_2]/(nmol\ mol^{-1})$ during daytime and $[NO]/(nmol\ mol^{-1}) = ([CO]/(nmol\ mol^{-1}) -$
$105)/30.9 - [NO_2]/(nmol\ mol^{-1})$ at night.
By using Eq. (2), the relative importance of nocturnal formation pathways ($f_{R8} + f_{R9} + f_{R10}$) can be written as Eq. (4):
$f_{R8} + f_{R9} + f_{R10} = \dfrac{f_{R9}}{2} + \dfrac{\Delta^{17}O(NO_3^-)}{14\text{‰}} - 1.8\alpha$     (4)
Eq. (4) suggests that the relative importance of nocturnal pathways is solely a function of the assumption of $f_{R9}$ at given
$\Delta^{17}O(NO_3^-)$ and $\alpha$. Since $f_{R9}, f_{R8} + f_{R10}$ and $f_{R8} + f_{R9} + f_{R10}$ should be in the range of $0 - 1$ all the time, $f_{R9}$ is further limited to
meet Eq. (5):
$f_{R9} \begin{cases} > 0 \\ < \min(1, \dfrac{\Delta^{17}O(NO_3^-)}{7\text{‰}} - 3.6\alpha, 2 + 3.6\alpha - \dfrac{\Delta^{17}O(NO_3^-)}{7\text{‰}}) \end{cases}$     (5)
We estimated the relative importance of nocturnal pathways ($f_{R8} + f_{R9} + f_{R10}$) by using concentration-weighted
$\Delta^{17}O(NO_3^-)$ observations and production rate weighted $\alpha$ in PD of each haze event rather than each sample due to the
lifetime of atmospheric nitrate is typically on the order of day (Liang et al., 1998), larger than our sampling collection
interval.
**2.4 Simulation of surface $N_2O_5$ and $NO_3$ radical**
To see whether the relative importance of nocturnal pathways constrained by $\Delta^{17}O(NO_3^-)$ can be reproduced by models,
we use the standard Master Chemical Mechanism (MCM, version 3.3, http://mcm.leeds.ac.uk/) to simulate the mixing
ratios of surface $N_2O_5$ and $NO_3$ radical during our sampling period. The input for this modeling work includes: (i) 1
h-averaged mixing ratios of observed surface CO, $NO_2$, $SO_2$ and $O_3$ and estimated NO (see Sect. 2.3), (ii) observed RH and
$T$, and (iii) the mixing ratios of organic compounds from the literatures (Table S1) (Wang et al., 2001; Wu et al., 2016; Rao et
al., 2016).

## 3 Results and Discussion

### 3.1 Overview of observations in Beijing haze

Figure 3 describes general characteristics of haze events during our observations. The 12h-averaged $PM_{2.5}$ concentrations, corresponding with filter samples, varied from 16 to 323 µg m$^{-3}$ with a mean of (141±88 (1SD)) µg m$^{-3}$. In comparison, the Grade II of NAAQS (National Ambient Air Quality Standard) in China is 75 µg m$^{-3}$ for daily $PM_{2.5}$. The $NO_3^-$ concentrations present similar trends with $PM_{2.5}$ levels (Fig. 3a), ranged from 0.3 to 106.7 µg m$^{-3}$ with a mean of (6.1±5.3) µg m$^{-3}$ in non-polluted days (NPD, $PM_{2.5} < 75$ µg m$^{-3}$) and (48.4±24.7) µg m$^{-3}$ in polluted days (PD, $PM_{2.5} \geq 75$ µg m$^{-3}$). Correspondingly, the nitrogen oxidation ratio (NOR, which equals to $NO_3^-$ molar concentration divided by the sum of $NO_3^-$ and $NO_2$ molar concentration), a proxy for secondary transformation of nitrate (Sun et al., 2006), increased from a mean of 0.09±0.05 in NPD to 0.31±0.10 in PD (Fig. 3b). In residential heating season (Case III – V in November 2014 – January 2015, Fig. 3b), $Cl^-$ concentrations present similar trends with $NO_3^-$ levels, increased from (0.6±1.0) µg m$^{-3}$ in NPD to (7.9±4.8) µg m$^{-3}$ in PD. However, during Case I – II in October 2014, $Cl^-$ concentrations were (3.5±1.6) µg m$^{-3}$ in NPD and (3.5±1.9) µg m$^{-3}$ in PD, showing no significant difference at 0.01 level (t-test). Throughout our observational period, the visibility decreased from (11.4±6.7) km in NPD to (3.1±1.8) km in PD (Fig. 3c) while relative humidity (RH) increased from (37±12) % in NPD to (62±12) % in PD (Fig. 3d).

$\Delta^{17}O(NO_3^-)$ ranged from 27.5 ‰ to 33.9 ‰ with the mean of (29.1±1.3) ‰ in NPD and (31.0±1.7) ‰ in PD (Fig. 3c). Our observed $\Delta^{17}O(NO_3^-)$ is in the range of aerosol $\Delta^{17}O(NO_3^-)$ reported in literatures (Table 3) and similar to wet deposition $\Delta^{17}O(NO_3^-)$ observed in East Asia (Nelson et al., 2018; Tsunogai et al., 2016; Tsunogai et al., 2010). All our observed $\Delta^{17}O(NO_3^-)$ values, no matter daytime sample (08:00 – 20:00) or nighttime sample (20:00 – 08:00), are larger than 25 ‰, the maximum of $\Delta^{17}O(NO_3^-)$ that can be produced via $NO_2 + OH$ and $NO_2 + H_2O$ (Table 1) at the assumption of bulk $\Delta^{17}O(O_3)$ = 26 ‰ (Ishino et al., 2017; Vicars and Savarino, 2014). This directly suggests nocturnal formation pathways ($N_2O_5$ + $H_2O/Cl^-$ and $NO_3$ + HC) must contribute to all the sampled nitrate. Given the lifetime of atmospheric nitrate is typically larger than our sampling collection interval (Vicars et al., 2013), each of our samples is expected to reflect both daytime and nocturnal nitrate production. Not surprisingly, $\Delta^{17}O(NO_3^-)$ mean of daytime and nighttime samples is (30.3±1.5) ‰ and (30.9±2.1) ‰, respectively, showing no significant difference at 0.01 level (t-test).

$\delta^{15}N(NO_3^-)$ in our observation varied from –2.5 ‰ to 19.2 ‰ with a mean of (7.4±6.8) ‰, which is in the range of $\delta^{15}N(NO_3^-)$ observed from rainwater in Beijing, China (Zhang et al., 2008) and similar to $\delta^{15}N(NO_3^-)$ values observed from aerosols in Germany (Freyer, 1991). Figure 3d shows that $\delta^{15}N(NO_3^-)$ varies largely in October 2014. The mean $\delta^{15}N(NO_3^-)$ varied from (0.4±1.5) ‰ in 08:00 Oct. 18 – 08:00 Oct. 21 to (10.7±1.4) ‰ in 08:00 Oct. 21 – 08:00 Oct. 23 and then decreased to (–0.9±2.1) ‰ in 08:00 Oct. 23 – 08:00 Oct. 26, which corresponds to $PM_{2.5}$ concentrations being 155±63,

$57 \pm 19$ and $(188 \pm 51)$ μg m$^{-3}$ respectively. However, during residential heating season, relatively high $\delta^{15}$N(NO$_3^-$) (7.6 –
19.2 ‰) were always observed both in NPD and PD. This may be related to the high NO$_X$ emission from coal combustion in
north China (Wang et al., 2012; Lin, 2012; Zhang et al., 2007).

## 3.2 Relationships between $\Delta^{17}$O(NO$_3^-$) and other data

Figure 4 presents the relationships between $\Delta^{17}$O(NO$_3^-$) and NO$_3^-$ concentrations, PM$_{2.5}$ concentrations, NOR, visibility,
RH and $\delta^{15}$N(NO$_3^-$). $\Delta^{17}$O(NO$_3^-$) shows a positive correlation with NO$_3^-$ concentrations when NO$_3^-$ < 50 μg m$^{-3}$ (r = 0.81, p
< 0.01). Similarly, $\Delta^{17}$O(NO$_3^-$) shows a positive correlation with PM$_{2.5}$ concentration in Fig. 4b and NOR in Fig. 4c when
NO$_3^-$ < 50 μg m$^{-3}$ (r = 0.71 and r = 0.80, p < 0.01, respectively). Figure 4d shows that $\Delta^{17}$O(NO$_3^-$) is negative correlated with
visibility in general (r = –0.66, p < 0.01). The significant decrease of visibility will largely reduce surface radiation and
thereby OH mixing ratios (Zheng et al., 2015b), which is unfavorable for nitrate production via NO$_2$ + OH pathway. Since
NO$_2$ + OH pathway produces low $\Delta^{17}$O(NO$_3^-$) (Table 1), the decreased importance of NO$_2$ + OH pathway will conversely
increase $\Delta^{17}$O(NO$_3^-$). While the raise of RH accompanying the large increase of PM$_{2.5}$ favors nitrate production via
heterogeneous uptake of gases, e.g., N$_2$O$_5$ (Zheng et al., 2015b; Zheng et al., 2015a) and heterogeneous uptake of N$_2$O$_5$
produces relative high $\Delta^{17}$O(NO$_3^-$) (Table 1), the enhanced heterogeneous uptake of N$_2$O$_5$ will increase $\Delta^{17}$O(NO$_3^-$) too.
Therefore, the decrease of importance of NO$_2$ + OH and the increase of importance of heterogeneous uptake of N$_2$O$_5$ should
be responsible for the positive correlation between $\Delta^{17}$O(NO$_3^-$) and NO$_3^-$ concentrations. In addition, for samples with NO$_3^-$ >
50 μg m$^{-3}$, visibility was always low with narrow variations (2.3 ± 1.0 km) and RH was always high with narrow range
(67 ± 7 %), which may be responsible for the relatively high $\Delta^{17}$O(NO$_3^-$) being observed (31.2 ± 1.7 ‰). Figure 4f shows that
$\Delta^{17}$O(NO$_3^-$) is not correlated with $\delta^{15}$N(NO$_3^-$).

## 3.3 Estimate of nocturnal formation pathways

Before estimating the relative importance of different nitrate formation pathways, we estimate the proportion of O$_3$
oxidation in NO$_2$ production rate, $\alpha$. The possible $\alpha$ range can be calculated based on observed $\Delta^{17}$O(NO$_3^-$). It can be
obtained from Table 1 that 25$\alpha$ ‰ < $\Delta^{17}$O(NO$_3^-$) < (25$\alpha$ + 14) ‰, so the lower limit of possible $\alpha$ is ($\Delta^{17}$O(NO$_3^-$) –
14 ‰)/25 ‰. And since $\Delta^{17}$O(NO$_3^-$) ≥ 27.5 ‰ in our observation, the higher limit of $\alpha$ is always 1 for all the samples. Figure
5 presents the possible range of calculated $\alpha$ based on $\Delta^{17}$O(NO$_3^-$). The calculated lower limit of $\alpha$ ranged from 0.56 to 0.81
with a mean of 0.68 ± 0.07, which directly suggests that O$_3$ oxidation played a dominated role in NO$_X$ cycling during Beijing
haze. To estimate the specific $\alpha$ value, chemical kinetics in Table 2 and Eq. (3) were used. Specific $\alpha$ is estimated to range
from 0.86 to 0.97 with a mean of (0.94 ± 0.03), which is in the possible range of $\alpha$ value calculated directly based on
$\Delta^{17}$O(NO$_3^-$) (Fig. 5) and close to the range of 0.85 – 1 determined in other mid-latitude areas (Michalski et al., 2003; Patris et
al., 2007).

Figure 6a shows the estimated relative importance of nocturnal formation pathways ($N_2O_5 + H_2O/Cl^-$ and $NO_3 + HC$)

during PD of each case on the basis of observed $\Delta^{17}O(NO_3^-)$. Possible fractional contribution of nocturnal formation
pathways ranges from 49 – 97 %, 58 – 100 %, 60 – 100 %, 45 – 90 % and 70 – 100 % in PD of Case I to V, respectively,
with a mean of 56 – 97 %. This directly implies that nocturnal chemistry dominates atmospheric nitrate production in Beijing
haze. This finding is consistent with the suggested importance of heterogeneous uptake of $N_2O_5$ during Beijing haze by
previous studies (Su et al., 2017; Wang et al., 2017b). The other pathways ($NO_2 + OH$ and $NO_2 + H_2O$) account for the
remaining fraction with a mean possible range of 3 – 44 %. Since $NO_2 + OH$ and $NO_2 + H_2O$ produces the same $\Delta^{17}O(NO_3^-)$
signature in our assumptions (Table 1), we cannot distinguish their fractional contribution barely from the observed
$\Delta^{17}O(NO_3^-)$ in the present study. However, the overall positive correlation between $\Delta^{17}O(NO_3^-)$ and RH (r = 0.55, p < 0.01,
Fig. 4e) suggests heterogeneous uptake of $NO_2$ should be less important than heterogeneous uptake of $N_2O_5$, otherwise, a
negative relationship between $\Delta^{17}O(NO_3^-)$ and RH is expected. Our calculations also suggest that the sum of possible
fractional contribution of $N_2O_5 + Cl^-$ and $NO_3 + HC$ is in the range of 0 – 49 %, 17 – 58 %, 20 – 60 %, 0 – 45 % and 41 – 70 %
in PD of Case I to V, respectively, with a mean of 16 – 56 % (Table 4), which emphasizes that $N_2O_5 + Cl^-$ and $NO_3 + HC$
played a non-ignorable role in nitrate production during Beijing haze. Due to that $N_2O_5 + Cl^-$ and $NO_3 + HC$ produce the
same $\Delta^{17}O(NO_3^-)$ in our assumptions (Table 1), we cannot distinguish their fractional contribution barely from the observed
$\Delta^{17}O(NO_3^-)$ in this study, either. However, $NO_3 + HC$ should be minor for nitrate production. For example, 3D modelling
work of Alexander et al. (2009) suggests $NO_3 + HC$ pathway only accounts for 4 % of global tropospheric nitrate production
annually on average, and Michalski et al. (2003) found that $NO_3 + HC$ pathway contributes 1 – 10 % to nitrate production on
the basis of an annual observation at La Jolla, California, with low values in winter. Therefore, in addition to $NO_3 + HC$,
$N_2O_5 + Cl^-$ is likely to also contribute to nitrate production during haze in Beijing. Supportively, the concentrations of $Cl^-$ is
as high as (5.5±4.1) µg m$^{-3}$ during PD of all the cases in our observation and the mixing ratios of $ClNO_2$, an indicator of
$N_2O_5 + Cl^-$ pathway, reached up to 2.9 nmol mol$^{-1}$ during a summer observation in suburban Beijing (Wang et al., 2018b)
and reached up to 5.0 nmol mol$^{-1}$ in a modelling work in summer rural Beijing (Wang et al., 2017c).

Figure 6b presents the simulated mixing ratios of surface $N_2O_5$ and $NO_3$ radical during our observational period by

using the box model MCM. The 12h averaged mixing ratios of simulated $N_2O_5$ ranged from 3 to 649 pmol mol$^{-1}$ while
simulated $NO_3$ radical ranged from 0 to 27 pmol mol$^{-1}$. In comparison, previous observations in Beijing suggest 5s averaged
$N_2O_5$ can be as high as 1.3 nmol mol$^{-1}$ and 30 min averaged $NO_3$ radical can be as high as 38 pmol mol$^{-1}$ with large
day-to-day variability (Wang et al., 2017b; Wang et al., 2015). During Case I and II in October, simulated $N_2O_5$ and $NO_3$
radical present similar trends with the observed $NO_3^-$ and remain relatively high during PD (346±128 pmol mol$^{-1}$ and 9±7
pmol mol$^{-1}$, respectively, Fig. 6b), which supports the dominant role of nocturnal formation pathways suggested by
$\Delta^{17}O(NO_3^-)$. However, during Case III – V in residential heating season, the simulated surface mixing ratios of $N_2O_5$ and
$NO_3$ radical remain relatively low during PD ($63\pm80$ pmol $mol^{-1}$ and $< 1$ pmol $mol^{-1}$, respectively, Fig. 6b), which seems to
be inconsistent with $\Delta^{17}O(NO_3^-)$ observations. We note that a recent study suggests that heterogeneous uptake of $N_2O_5$ is
negligible at surface but larger at higher altitudes (e.g., $> 150$ m) during winter haze in Beijing (Wang et al., 2018a). So
during PD of Case III – V in our observational period, large nitrate production via heterogeneous uptake of $N_2O_5$ may occur
aloft rather than at surface, which leads to the dominant role of nocturnal formation pathways as suggested by $\Delta^{17}O(NO_3^-)$.
**4 Conclusions**
We report the first observation of isotopic composition ($\Delta^{17}O$ and $\delta^{15}N$) of atmospheric nitrate in Beijing haze. The
observed $\Delta^{17}O(NO_3^-)$ ranged from 27.5 ‰ to 33.9 ‰ with a mean of ($30.6\pm1.8$) ‰. $\delta^{15}N(NO_3^-)$ ranged largely from −2.5 ‰
to 19.2 ‰ with a mean of ($7.4\pm6.8$) ‰. When $NO_3^-$ is $< 50$ μg $m^{-3}$, a positive correlation was observed between $\Delta^{17}O(NO_3^-)$
and $NO_3^-$ concentration ($r = 0.81$, $p < 0.01$). This is likely to result from the variation of relative importance of different
nitrate formation pathway. Calculations with the constraint of $\Delta^{17}O(NO_3^-)$ suggest that nocturnal pathways ($N_2O_5 + H_2O/Cl^-$
and $NO_3 + HC$) dominated nitrate production during polluted days ($PM_{2.5} \geq 75$ μg $m^{-3}$), with the mean possible contribution
of 56 – 97 %. $\Delta^{17}O(NO_3^-)$ also indicates that $O_3$ dominated NO oxidation during Beijing haze.
**Supplementary Materials**
**Figure S1.** The diurnal differences of observed $NO_2$, CO and $O_3$ and calculated NO, $HO_2$ and $RO_2$ during our sampling
periods.
**Table S1.** The input of organic compounds for MCM model (nmol $mol^{-1}$).
**Data availability**
All data needed to draw the conclusions are present in the main text and/or the Supplementary Materials. For additional
data, please contact the corresponding author (zqxie@ustc.edu.cn).
**Author contributions**
Z.Q.X. conceived this study. P.Z.H. conducted isotope measurements. P.Z.H., X.Y.C, S.D.F., H.C.Z., H. K. performed
the field experiments and ion measurements. P.Z.H., Z.Q.X., X.W.Y. interpreted the data. C.L. contributed to the field
observation support. P.Z.H. wrote the manuscript with Z.Q.X. inputs. All authors involved the discussion and revision.
**Competing interests**

The authors declare no competing interests.

**Acknowledgments**

This work was supported by the National Key Project of MOST (2016YFC0203302), NSFC (91544013), the Key

Project of CAS (KJZD-EW-TZ-G06-01) and the Atmospheric Pollution Control of the Prime Minister (DQGG0104). We
gratefully thank staffs of IsoLab at UW for their technical support, Becky Alexander and Lei Geng for helpful discussions.

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

**Figures and Tables**

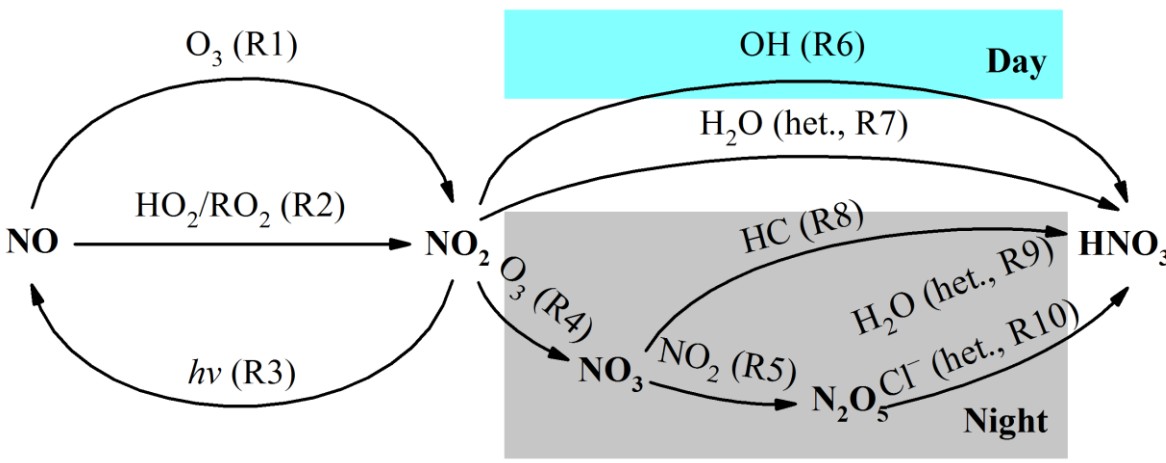


**Figure 1.** Simplified schematic of the main nitrate formation pathways in urban air. "het." means heterogeneous reactions on
aerosols.

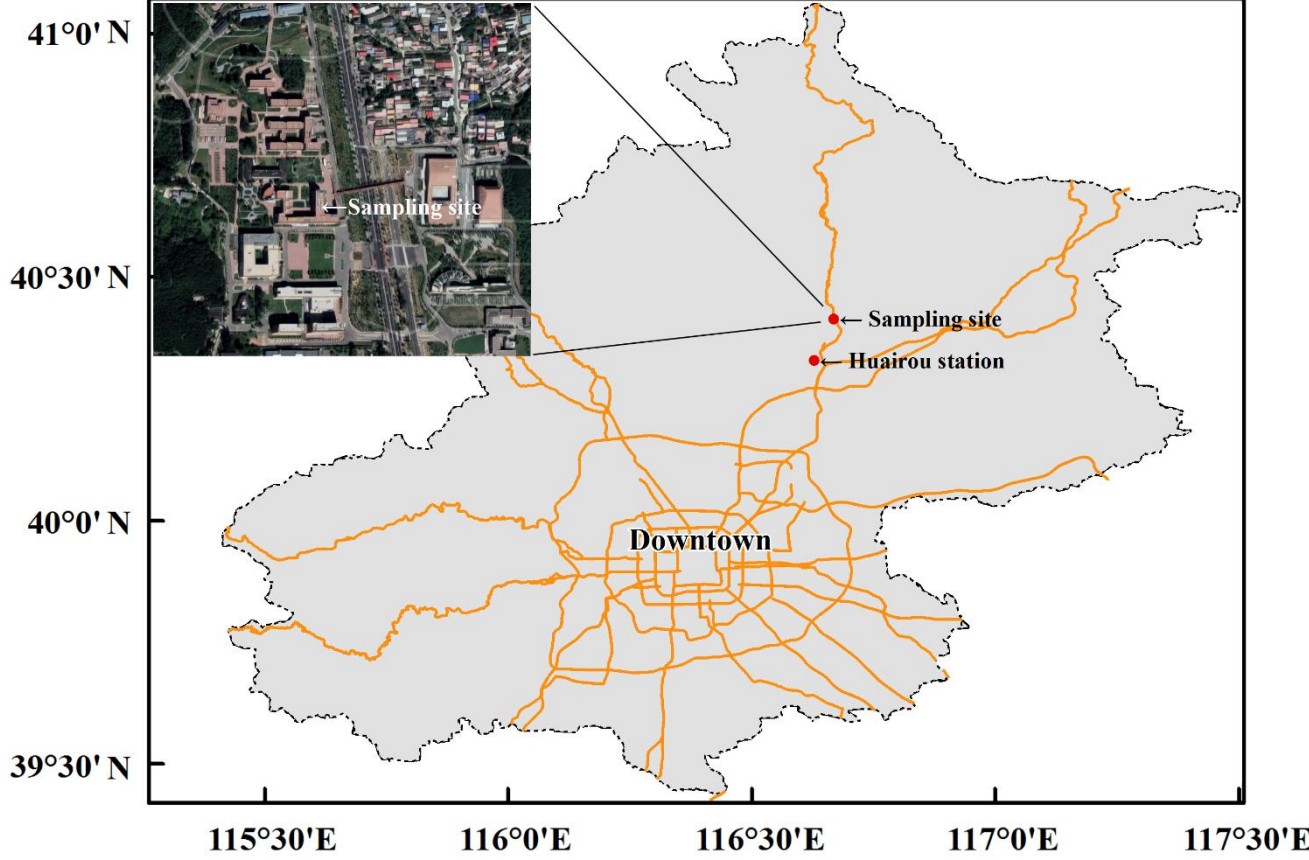


**Figure 2.** A brief map of sampling site in Beijing. The map scale of base map is 1:1250000. Huairou station is set by Beijing
Municipal Environmental Monitoring Center, where hourly $PM_{2.5}$, $SO_2$, CO, $NO_2$ and $O_3$ were observed.

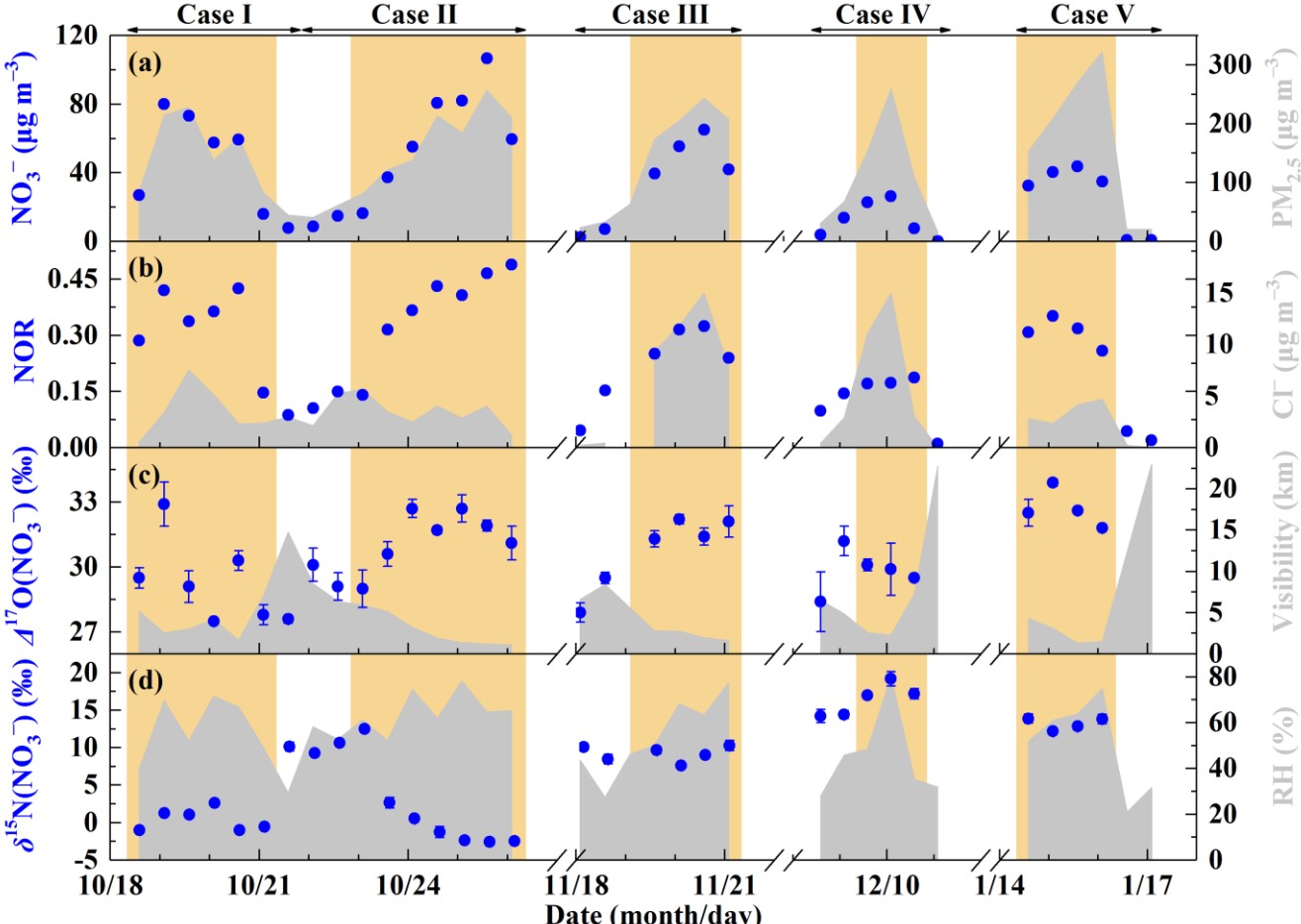


**Figure 3.** General characteristics of haze events in Beijing (October 2014 – January 2015). **(a)** Time series of $PM_{2.5}$ and
$NO_3^-$ concentrations. **(b)** Time series of nitrogen oxidation ratio (NOR, which equals to $NO_3^-$ molar concentration divided by
the sum of $NO_3^-$ and $NO_2$ molar concentration) and $Cl^-$ concentrations. **(c)** Time series of $\Delta^{17}O(NO_3^-)$ and visibility. **(d)** Time
series of $\delta^{15}N(NO_3^-)$ and relative humidity (RH). The error bars in (c) and (d) are $\pm 1\sigma$ of replicate measurements (n = 3) of
each sample. The khaki shaded area indicates polluted days (PD, $PM_{2.5} \geq 75\ \mu g\ m^{-3}$).

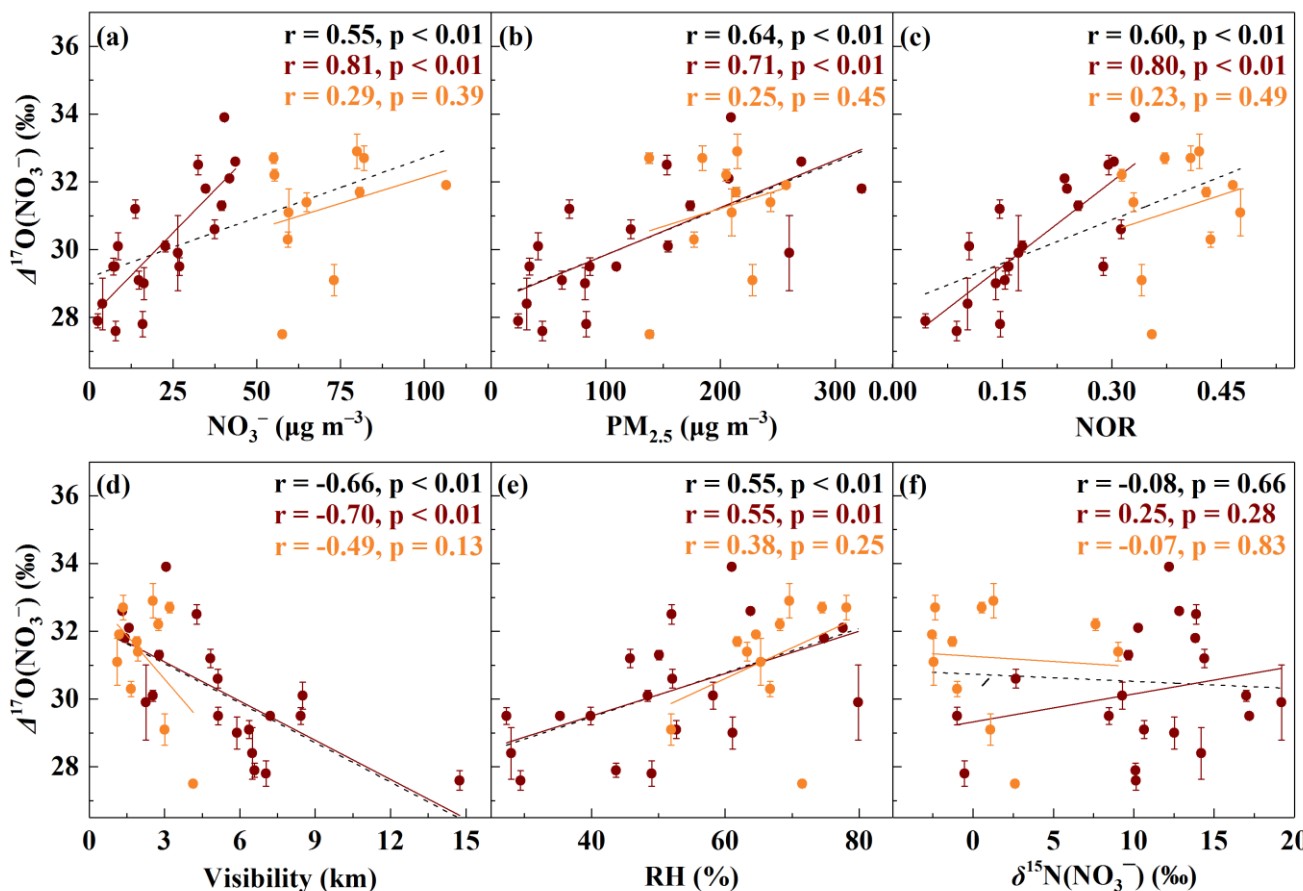


**Figure 4.** Relationships between $\Delta^{17}O(NO_3^-)$ and other parameters. The relationship between $\Delta^{17}O(NO_3^-)$ and $NO_3^-$ concentrations (**a**), $PM_{2.5}$ concentrations (**b**), nitrogen oxidation ratio (NOR, **c**), visibility (**d**), relative humidity (RH, **e**) and $\delta^{15}N(NO_3^-)$ (**f**). The dark red dots are samples with $NO_3^- < 50$ μg m$^{-3}$ and the orange dots are samples with $NO_3^- > 50$ μg m$^{-3}$. The black dash lines are linear least-squares fitting lines for all samples, the dark red solid lines are linear least-squares fitting lines for samples with $NO_3^- < 50$ μg m$^{-3}$ and the orange solid lines are linear least-squares fitting lines for samples with $NO_3^- > 50$ μg m$^{-3}$. The error bars are $\pm 1\sigma$ of replicate measurements of each sample.

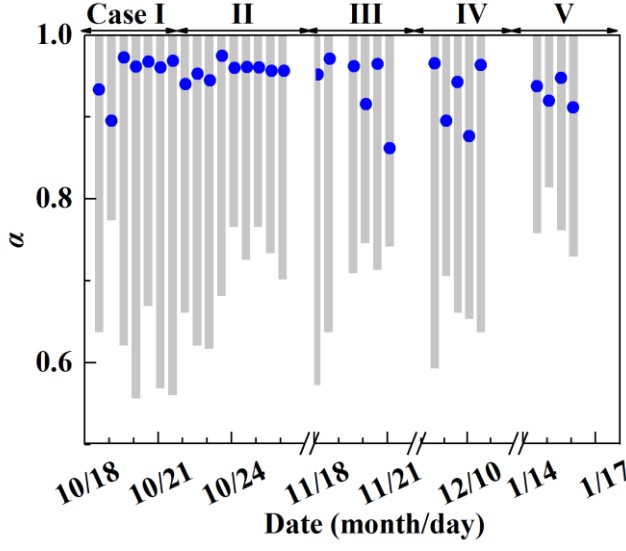


**Figure 5.** Estimate of the proportion of $O_3$ oxidation in $NO_X$ cycling, $\alpha$. The gray column represents possible $\alpha$ range

determined by $\Delta^{17}O(NO_3^-)$. The blue dot represents specific $\alpha$ value calculated by Eq. (3).

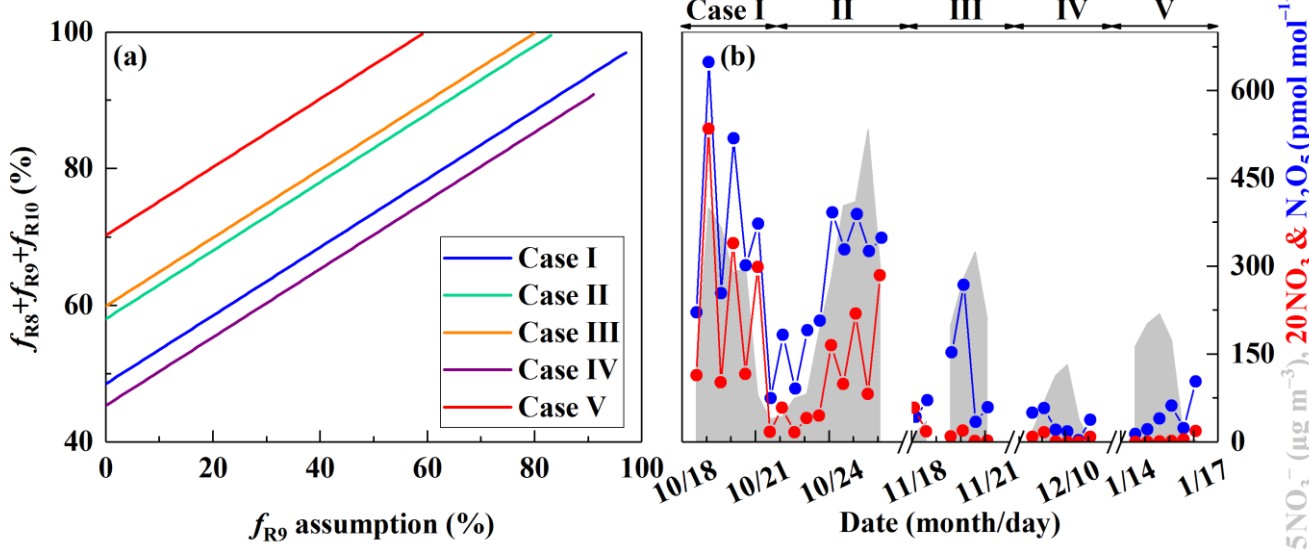


**Figure 6.** Estimate of the nocturnal formation pathways. The estimated relative importance of nocturnal formation pathways
($f_{R8} + f_{R9} + f_{R10}$) during PD of each case on the basis of observed $\Delta^{17}O(NO_3^-)$ (See Sect. 2.3, **a**) and the simulated mixing
ratios of $N_2O_5$ and $NO_3$ radical by MCM (**b**). R8, R9 and R10 in (a) represents $NO_3 + HC$, $N_2O_5 + H_2O$ and $N_2O_5 + Cl^-$
pathway, respectively.
**Table 1.** Isotope assumptions of different nitrate formation pathways.

| No. | Reaction | $\Delta^{17}O$ of product Expression | Value (‰) [a] | Reference |
|---|---|---|---|---|
| R1 | $NO + O_3 \rightarrow NO_2 + O_2$ | $\Delta^{17}O(NO_2) = 1.18 \times \Delta^{17}O(O_3) + 6.6\ ‰$ | 37 | (Savarino et al., 2008) |
| R2 | $NO + HO_2/RO_2 \rightarrow NO_2 + OH/RO$ | $\Delta^{17}O(NO_2) = 0.0$ | 0.0 | (Sofen et al., 2014) |
| R4 | $NO_2 + O_3 \rightarrow NO_3 + O_2$ | $\Delta^{17}O(NO_3) =$ $\frac{2}{3}\Delta^{17}O(NO_2) + \frac{1}{3}(1.23 \times \Delta^{17}O(O_3) + 9.0\ ‰)$ | $25\alpha + 14$ | (Berhanu et al., 2012) |
| R5 | $NO_2 + NO_3 \rightarrow N_2O_5$ | $\Delta^{17}O(N_2O_5) = \frac{2}{5}\Delta^{17}O(NO_2) + \frac{3}{5}\Delta^{17}O(NO_3)$ | $30\alpha + 8$ | (Sofen et al., 2014) |
| R6 | $NO_2 + OH \rightarrow HNO_3$ | $\Delta^{17}O(NO_3^-) = \frac{2}{3}\Delta^{17}O(NO_2)$ | $25\alpha$ | (Sofen et al., 2014) |
| R7 | $2NO_2 + H_2O \rightarrow HNO_3 + HNO_2$ | $\Delta^{17}O(NO_3^-) = \frac{2}{3}\Delta^{17}O(NO_2)$ | $25\alpha$ [b] | |
| R8 | $NO_3 + HC \rightarrow HNO_3 + products$ | $\Delta^{17}O(NO_3^-) = \Delta^{17}O(NO_3)$ | $25\alpha + 14$ | (Sofen et al., 2014) |
| R9 | $N_2O_5 + H_2O \rightarrow 2HNO_3$ | $\Delta^{17}O(NO_3^-) = \frac{5}{6}\Delta^{17}O(N_2O_5)$ | $25\alpha + 7$ | (Sofen et al., 2014) |
| R10 | $N_2O_5 + Cl^- \rightarrow HNO_3 + ClNO_2$ | $\Delta^{17}O(NO_3^-) = \Delta^{17}O(NO_3)$ | $25\alpha + 14$ [c] | |

[a] The values are calculated on assumptions that bulk $\Delta^{17}O(O_3) = 26$ ‰ (Vicars and Savarino, 2014; Ishino et al., 2017) and
$\Delta^{17}O(HO_2/RO_2) = 0$ ‰. $\Delta^{17}O(RO_2)$ is equal to 0 ‰ in the troposphere (Morin et al., 2011), in contrast, observations suggest
$\Delta^{17}O(HO_2) = 1 - 2$ ‰ (Savarino and Thiemens, 1999). However, the difference in calculated $\Delta^{17}O(NO_3^-)$ between assuming
$\Delta^{17}O(HO_2) = 0$ ‰ and $\Delta^{17}O(HO_2) = 2$ ‰ is negligible in this study (< 0.1 ‰). And the assumption that $\Delta^{17}O(HO_2) = 0$ ‰
simplifies calculations and is also consistent with previous studies (Michalski et al., 2003; Alexander et al., 2009; Morin et
al., 2008; Kunasek et al., 2008; Sofen et al., 2014). $\alpha$ is the proportion of $O_3$ oxidation in $NO_2$ production rate, calculated by
Eq. (3).
[b] Previous studies suggest that in R7 one oxygen atom of $NO_3^-$ is from $H_2O$ and the other two are from $NO_2$ (Li et al., 2010;
Cheung et al., 2000; Goodman et al., 1999), which will result in $\Delta^{17}O(NO_3^-) = 2/3\Delta^{17}O(NO_2)$.
[c] R4 and R5 suggest that the central oxygen atom of $N_2O_5$ ($O_2N$-$O$-$NO_2$) is from $NO_3$ radical ($O$-$NO_2$) with $\Delta^{17}O$ (‰) =
$1.23 \times \Delta^{17}O(O_3) + 9.0$ ‰. R10 is suggested to occur via $O_2N$-$O$-$NO_2$ (aq) $\leftrightarrow NO_2^+ + NO_3^-$ and the following $NO_2^+ + Cl^- \rightarrow$
$ClNO_2$ (Bertram and Thornton, 2009), so $\Delta^{17}O(NO_3^-) = 1/3(1.23 \times \Delta^{17}O(O_3) + 9.0$ ‰$) + 2/3\Delta^{17}O(NO_2) = \Delta^{17}O(NO_3)$.
**Table 2.** Reaction expressions for different $NO_2$ production pathways.

| No. | Reaction | Rate expression | Rate constant (cm³ molecule⁻¹ s⁻¹) | Reference |
|---|---|---|---|---|
| R1 | $NO + O_3 \rightarrow NO_2 + O_2$ | $k_{R1}[NO][O_3]$ | $k_{R1}=3.0\times10^{-12}\times e^{(-1500/T)}$ | (Burkholder et al., 2015) |
| R2a | $NO + HO_2 \rightarrow NO_2 + OH$ | $k_{2Ra}[NO][HO_2]$ | $k_{2Ra}=3.3\times10^{-12}\times e^{(270/T)}$ | (Burkholder et al., 2015) |
| R2b | $NO + RO_2 \rightarrow NO_2 + RO$ | $k_{2Rb}[NO][RO_2]$ | $k_{2Rb} = k_{2Ra}$ | (Burkholder et al., 2015; Kunasek et al., 2008) |

**Table 3.** Atmospheric $\Delta^{17}O(NO_3^-)$ in aerosols obtained from the literature and this study.

| Sample location | Sample period | Collection interval | $\Delta^{17}O$ (‰) range | Reference |
|---|---|---|---|---|
| Huairou, Beijing (40.41 °N, 116.68 °E) | October 2014 – January 2015 | 12 h | 27.5 – 33.9 (30.6 ± 1.8) | This study |
| Trinidad Head, California (41.0 °N, 124.2 °W) | April – May 2002 | 1 – 4 days | 20.1 – 27.5 | (Patris et al., 2007) |
| La Jolla, California (32.7 °N, 117.2 °W) | March 1997 – April 1998 | 3 days | 20 – 30.8 | (Michalski et al., 2003) |
| Mt. Lulin, Taiwan (23.5 °N, 120.9 °E) | January – December 2010 | 1 day | 2.7 – 31.4 (17 ± 7) | (Guha et al., 2017) |
| Cape Verde Island (16.9 °N, 24.9 °W) | July 2007 – May 2008 | 2 – 3 days | 25.5 – 31.3 | (Savarino et al., 2013) |
| Cruise in costal California (32.8 °N – 38.6 °N) | May – June 2010 | 2 – 22 h | 19.0 – 29.2 (24.1 ± 2.2) | (Vicars et al., 2013) |
| Cruise from 65 °S to 79 °N | September – October 2006 | 1 – 4 days | Non-polar: | (Morin et al., 2009) |

| | April – May 2007 | | 24 – 33 | |
| | February – April 2006 | | Polar: 35 ±2 | |
| Alert, Nunavut | March – May 2004 | 3 – 4 days | 29 – 35 | (Morin et al., 2007b) |
| (82.5 °N, 62.3 °W) | | | (32.7 ±1.8) | |
| Barrow, Alaska | March 2005 | 1 day | 26 – 36 | (Morin et al., 2007a) |
| (71.3 °N, 156.9 °W) | | | | |
| Dumont d'Urville, Antarctic | January – December 2001 | 10 – 15 days | 20.0 – 43.1 | (Savarino et al., 2007) |
| (66.7 °S, 140.0 °E) | | | | |
| Dumont d'Urville, Antarctic | January 2011 – January 2012 | 7 days | 23.0 – 41.9 | (Ishino et al., 2017) |
| (66.7 °S, 140.0 °E) | | | | |

**Table 4** The possible range of fractional contribution of different nitrate formation pathways during PD of each case
estimated on the basis of observed $\varDelta^{17}O(NO_3^-)$ [a].

| PD of Case | $f_{R9}$ assumption (%) | $f_{R8} + f_{R9} + f_{R10}$ (%) | $f_{R8} + f_{R10}$ (%) | $f_{R6} + f_{R7}$ (%) |
|---|---|---|---|---|
| I | 0 – 97 | 49 – 97 | 0 – 49 | 3 – 51 |
| II | 0 – 83 | 58 – 100 | 17 – 58 | 0 – 42 |
| III | 0 – 80 | 60 – 100 | 20 – 60 | 0 – 40 |
| IV | 0 – 90 | 45 – 90 | 0 – 45 | 10 – 55 |
| V | 0 – 59 | 70 – 100 | 41 – 70 | 0 – 30 |
| Average | 0 – 82 | 56 – 97 | 16 – 56 | 3 – 44 |

[a] R6, R7, R8, R9 and R10 is respectively $NO_2 + OH$, $NO_2 + H_2O$, $NO_3 + HC$, $N_2O_5 + H_2O$ and $N_2O_5 + Cl^-$ pathway.