# Peer review of "Atmospheric $\Delta^{17}O(NO_3^-)$ reveals nocturnal chemistry dominates nitrate production"

_Atmospheric Chemistry and Physics, 2018_

## Referee Comment (RC1) · Anonymous Referee #2 · 1 Jun 2018

Review of "Atmospheric $\Delta^{17}O(NO_3^-)$ reveals nocturnal chemistry dominates nitrate production in Beijing haze" by He et al.

In this study, the authors investigated the formation pathways of nitrate based on $\Delta^{17}O(NO_3^-)$ and $\delta^{15}N(NO_3^-)$. The authors concluded that nocturnal pathways ($N_2O_5 + H_2O$ and $NO_3$ radical + hydrocarbon) dominated the nitrate production during polluted days. Measuring the isotopic composition is an important, but underutilized approach to reveal the sources and formation pathways of atmospheric species. This study brings new insights into the nitrate sources during polluted days in Beijing. Overall, the interpretation of results is sound. However, there is room for improving the discussions. While I suggest publication after major revision, I hope that the authors will consider the following comments to make the manuscript more readable and hopefully more impactful.

Major Comments

1.      "Nitrate" is not clearly defined in the manuscript. Based on reactions in Table 1, "nitrate" refers to HNO3. However, in method section, filter-extracted $NO_3^-$ ion is analyzed. Is the implicit assumption that there is no isotope fractionation from $HNO_3$ to $NO_3^-$? Please clarify. In the literature, "nitrate" sometimes includes both inorganic nitrate (e.g., $NH_4NO_3$) and organic nitrate (e.g., isoprene hydroxyl nitrate). Please clarify if organic nitrate is included in the analysis of this study? In other words, can organic nitrate be analyzed by the bacterial denitrifier method?

2.      Correlation between $\Delta^{17}O(NO_3^-)$ and $[NO_3^-]$. It is plausible that the positive correlation is caused by that nocturnal pathways contribute more the $[NO_3^-]$. However, how to explain that the correlation is degraded when $[NO_3^-]$ is $> 50$ $\mu g$ $m^{-3}$? Does it suggest that when $[NO_3^-]$ is high, $NO_3^-$ is not from nocturnal pathways?

3.      Section 3.4.4 is confusing. If coal combustion is the major contributor to NOx and coal combustion has the largest $\delta^{15}N(NO_3^-)$, why is the $\delta^{15}N(NO_3^-)$ very low (i.e., mostly ~0) in October?

4.      Many calculations are not clearly described. For example, line 214-217, it is not clear how these fractional values are calculated. Line 277, how is $[\delta^{15}N(NO_2)- \delta^{15}N(NO_x)]$ calculated? On a related note, what is the rationale to correlate $\delta^{15}N(NO_3^-)$ with $[\delta^{15}N(NO_2)- \delta^{15}N(NO_x)]$?

Minor Comments

1.      Line 118-126. Show the estimated diurnal trends in the SI.

2.      Section 2.4. Discuss the purpose of using MCM estimation.

3.      Line 194-203. The authors used two methods to estimate the alpha value. These two methods should be compared and the discrepancies should be discussed.

4.      There are many gramma errors in the manuscript. For example, line 249, add "that" after "suggest". Sentences from line 304 to 306 and from line 263-267 have many gramma errors. These two sentences are too long and should be broken down. The authors should check throughout the manuscript.

---

## Referee Comment (RC2) · J. Rudolph (Referee) · 25 Jun 2018

The paper presents an interesting example for the use of isotope ratio measurements to gain insight into complex atmospheric reaction systems, here the formation of nitric acid and nitrate from NOx. Overall the paper is well written, the experimental work and interpretation solid and the subject (particle formation by oxidation of primary atmospheric pollutants is relevant for air quality. I also appreciate that the authors openly explain that isotope ratio studies in complex systems can only provide constraints (here given as range of possible contributions to nitrate formation) and that additional information is required to fully understand the magnitude of contributions from different individual reaction pathways. Consequently, I recommend publication although the authors need to address some questions and uncertainties in more detail before the

paper should be accepted for publication. i) May main concern is that the paper does not consider the photolysis of NO2 during daytime. Although this reaction is included in Figure 1 (R3), it is not considered in the excess oxygen calculation. During daytime the reaction sequence NO2+hv=>NO+O O+O2=O3 NO+O3=>NO2+O2 (R1) will result in a steady state which can (depending on photon flux and ozone concentration) be established within several minutes. This will result not only in an isotope exchange for N between NO and NO2 (Chapter 3.4.3) but also for O between NOx, O2 and O3. In contrast to this at night R1 is a one-way street. I do not know to which extent the daytime "recycling" of NO from NO2 photolysis will impact the excess oxygen ratio in NO2 and NO (and consequently in nitrate) or the 15N isotope ratio. Nevertheless, this is something that needs to be explained and discussed and potentially may change the interpretation of the isotope ratio measurements. ii) The authors use several approximations and comparisons with published results (e.g. for estimating NO, the contribution of specific pathways of nitrate formation etc.). The validity of applying these published results for this study will depend on pollution levels, degree of impact of local sources, contribution from processed polluted air masses and so on and therefore may nor be directly applicable to the cases studied here. This needs to be explained and discussed in more detail. iii) The various values (e.g. rate constants, excess isotope ratios in Table 2, estimates of [NO] from [CO]) used in the calculations will have uncertainties, which will add uncertainty to all quantitative results. This needs to evaluated in more detail. iv) Subchapter 3.4.1: Indeed, the impact of deposition on 15N is difficult to estimate. The argument that the impact of partitioning between gas and PM is minor since bot HNO3 and nitrate are collected on the filter is not convincing. Deposition rates for HNO3 and nitrate differ and will be highly variable depending on the situation. If the 15N isotope ratios for PM nitrate and gas phase HNO3 differ, differences in deposition rates will change the isotope ratio for the sum of HNO3 and nitrate. v) Chapter 3.4.3: This chapter neglects the NO+O3 and NO2+hv cycle (see above) Furthermore f NOx (in Eq. 6) is based on [NO] values calculated from measured [CO] and [NO2] and consequently the calculated values for ($\delta$15N(NO2)- $\delta$15N(NOx)) are in reality a

non-linear function of the [NO2] and [CO] concentrations. Thus Figure 7a is a plot of $\delta$15N(NO3-) versus a non-linear function of [NO2] and [CO]. Not sure how to interpret this, but obviously [NO2] and[CO] will vary for different sources with different 15N values. In order to be of value for the reader there needs a more detailed discussion than "should therefore be interpreted with the consideration of atmospheric contexts". The discussion of $\delta$15N(NO3-) should be combined into one chapter discussing the different factors that may influence $\delta$15N(NO3-). Due to the complexity of the various factors influencing $\delta$15N(NO3-) the attempt to discuss individual contributions separately does not work well.

A revised version considering these specific problems will merit publication.

Details:

General: Often a values are given as (xyz$\pm$abc), it is not always clear whether the $\pm$ indicates the error of the mean or the standard deviation.

Correlations: If I understand correctly, the authors present r and not r2. R values of 0.5 or so correspond to r2 of 0.25, a very weak correlation. These low r values need a more critical discussion of their meaning. It maybe that even a weak correlation has statistical validity. However, it has to be remembered that for r=0.5, r2=0.25, which means that only 25% of the observed variability can be explained by a linear dependence between dependent and independent variable.

The authors use "wine colored" in several figure captions. Dark red would be better.

53: . And once formed

76: Sampling site

78: Super site set by..

81: About 10 km to our sampling site

88, 94: Insoluble substances were filtered (removed by filtration?)

90: When determine the. . .

90: precision by our

95: which were decomposed from

110, 111 and other lines: is respectively

130: at the same time

133, 134: I assume weighted averages are meant. I understand the meaning and rational for concentration weighted oxygen excess, but I am not sure what production rate weighted means. $\alpha$ is a ratio with the total NO2 production rate in the denominator, consequently the production rate weighted average for $\alpha$ would be some kind of average for the nominator, that is k[NO][O3]. This requires more clarification and explanation.

164: samples

251: a small snow lasted for..

258: . . .it has been proposed that atmospheric nitrate that resulting from heterogeneous uptake of N. . ..

262: Don't present similar trends..

518:is set by 551: . And

---

## Referee Comment (RC3) · G. Michalski (Referee) · 3 Jul 2018

Line 114 it is unclear to what the coefficients 24.85 and 13.66 mean or where they are derived. As someone versed in the field, and some information on line 26, I can surmise this is the $\Delta^{17}O$ value NO2+OH pathway, but this is in no way clear to the non-specialist. There are host of assumptions that go into this number that are not explained and have uncertainties that are not being propagated through. Six points on this are

1. From the text there is the assumption that the $\Delta^{17}O$ of $O_3$ is essentially a fixed value of 26‰, which is by no means codified in the literature, despite the some who would hope so because it makes the data analysis less problematic.
2. Johnston et al. and Krankowsky et al. observed $O_3$ $\Delta^{17}O$ values that spanned 18.8‰ to 41‰ with a standard deviation of 4.8‰.
3. Two papers by the Savarino group using a different method arrive at values close to 26‰ with smaller variations of 1 and 1.6‰. Their Antarctic paper noted $O_3$ $\Delta^{17}O$ had " insignificant variation" 28 ‰ - 23 ‰, if one considers ~20% variation insignificant.
4. Lab experiments have clearly noted an $O_3$ $\Delta^{17}O$ temperature dependence.
5. NOx photochemical equilibrium experiments (Michalski et al., 2013) a higher terminal atoms value transfer and Vicars noted that $\Delta^{17}O(O_3)_{trans.}$ in the range of 38–44‰ fits data.
6. Even assuming a fixed value of $O_3$ $\Delta^{17}O$ value of 26‰, one cannot increase significant (24.85) digits by division/multiplication.

The authors should note these conflicting assumptions and how these assumptions would influence their interpretations of reaction pathways.

NO was derived CO mixing ratios derived from observations in winter Beijing (Lin et al., 2011). The correlation coefficients for this relationship are .76 and .82, which means there is some uncertainty in the derived NO. How would this impact the author's results

Line 200: "To estimate the specific $\alpha$ value, chemical kinetics in Table 2 and Eq. (3) were used. Specific $\alpha$ is estimated to range from 0.86 to 0.97 with a mean of (0.94±0.03)". The coefficients used to estimate HO2 has significant uncertainties (again r2= ~0.7) and the regression itself is are not universal but are valid for Tokyo. No discussion on whether this would hold in an extreme haze event in Beijing. Likewise the uncertainty of RO2 = 0.7HO2 must be significant and site specific. The validity of this assumption in the context of extreme haze needs to be discussed

"lifetime of atmospheric nitrate is typically on the order of days (Vicars et al., 2013)" I doubt that Vicars was the first to determine the lifetime of nitrate in the atmosphere. Further the lifetime is significantly dependent on precipitation frequency so if there was no rain during the collection period the lifetime of nitrate is significantly longer, though it does not change the authors point.

138 "We use the Master Chemical Mechanism " This requires an entire discussion section. MCM is a gas phase mechanism. Were heterogeneous reactions included? Based on what uptake scheme? How was aerosol surface area determined if that was part of the scheme? "1 h-averaged mixing ratios of observed surface CO, NO2, SO2 and O3 and estimated NO" what does this mean? Did you initialize the model with these mixing ratios? Or did you correct the model to match these hourly? Or did you run the model hourly? What length of spin-up do you use? How was photolysis adjusted to account for haze? This model predicts things like OH2, RO2,

NO…how does the model prediction compare with your estimation of these key compounds that we parameterized by your isotope scheme, but not measured? How does it predict things that were measured over time ($O_3$, NO2,)? This section was entirely too vague for anything useful to be inferred about the accuracy of predicted NO3 or $N_2O_5$ mixing ratios.

"variation of atmospheric $\delta^{15}N(NO3^-)$ can be interpreted by the following four processes (Vicars et al.,2013)" again please give credit where credit is due, Freyer used this scheme 20 years before Vicars to investigate 15N variations in atmospheric nitrate.

254 "The quartz filter used here is thought to collect both particulate nitrate and gaseous $HNO_3$" this statement needs better justification by citing filter pack studies. This is particularly true in Beijing where $NH_4NO_3$ is a major component of PM and loss by volatilization could also be occurring. Vicars, like myself (2003), limited this assumption to coastal sampling where seas salt buffering was present and noted that "the exact nature of the nitrate species collected during sampling using glass fiber filters has always been an area of some debate due primarily $NH_4NO_3$.

Isotopic fractionations associated with nitrate formation pathways. These (Photolysis and KIE effects in NOy) are largely unknown and the discussion should reflect that. Walters ab initio paper indicates IF equilibrium is dominant the more oxidized compounds should have higher 15N. Is this consistent with observations?

275 "Where K is the isotopic exchange constant of N between NO and NO2, which is temperature-dependent .." It is not clear if the authors are using temperature to calculate this daily, if so what temperature? Average? Day and night average? Clearly this equation is very dependent on fraction of NO2, which is based on NO estimations that also have uncertainty, which should be discussed and represented on the y-axis error bar on figure 7. That caption should emphasize theY data is not a measurement of the $\delta^{15}N$ of ambient NOx (Freyer, Walters) but a calculation. It would also seem that since the authors are presenting $\delta^{15}N$ in ‰, that the RHS of Eq, 6 will need to be multiplied by a factor of 1000.

279 "the correlation is better in residential heating season … especially in residential heating season. " mechanistic, why would this so? The authors seem to imply residential heating is promoting exchange when its likely NO/NO2 ratios. Was the a correlation between $\delta^{15}N$ and fNO2?

The exchange section should discuss in terms of Freyers and Walters et al. papers that measured $\delta^{15}N$ values of ambient NO2

"Influence of NOX emissions." This section could be greatly expanded, there has been a lot of recent work by the Elliot, Hastings, and Michalski groups of 15N sources. While coal maybe be dominant in the surrounding regions, automobiles and diesel trucks in Beijing must be significant, particularly during stagnant conditions. Is there a better N inventory for Being itself?

I did not see any discussion about any (or lack thereof) correlation between $\delta^{18}O$ $\Delta^{17}O$ and $\delta^{15}N$. If they are completely decoupled then that would argue for source effects, if there is some covariation, then exchange/chemistry could be the main process.

---

## Author Comment (AC1) · 7 Sep 2018

**Referee #1:**

**Comments:** In this study, the authors investigated the formation pathways of nitrate based on $\Delta^{17}O(NO_3^-)$ and $\delta^{15}N(NO_3^-)$. The authors concluded that nocturnal pathways ($N_2O_5 + H_2O$ and $NO_3$ radical + hydrocarbon) dominated the nitrate production during polluted days. Measuring the isotopic composition is an important, but underutilized approach to reveal the sources and formation pathways of atmospheric species. This study brings new insights into the nitrate sources during polluted days in Beijing. Overall, the interpretation of results is sound. However, there is room for improving the discussions. While I suggest publication after major revision, I hope that the authors will consider the following comments to make the manuscript more readable and hopefully more impactful.

*A: Thanks very much for your comments. We reply to your comments one by one as follows. One point needs to be addressed here is that we have removed section 3.4 from the manuscript due to that we are unable to explain the variations of $\delta^{15}N(NO_3^-)$ well so far.*

Major Comments

1. "Nitrate" is not clearly defined in the manuscript. Based on reactions in Table 1, "nitrate" refers to $HNO_3$. However, in method section, filter-extracted $NO_3^-$ ion is analyzed. Is the implicit assumption that there is no isotope fractionation from $HNO_3$ to $NO_3^-$? Please clarify. In the literature, "nitrate" sometimes includes both inorganic nitrate (e.g., $NH_4NO_3$) and organic nitrate (e.g., isoprene hydroxyl nitrate). Please clarify if organic nitrate is included in the analysis of this study? In other words, can organic nitrate be analyzed by the bacterial denitrifier method?

*A: Thanks for your comments. In this manuscript, atmospheric nitrate is defined as gas-phase $HNO_3$ plus particulate $NO_3^-$, which is the filter-extracted $NO_3^-$ ion analyzed by ion chromatography and is consistent with previous studies (e.g., (Vicars et al., 2013; Morin et al., 2009; Michalski et al., 2003; Alexander et al., 2009)). Once formed, the oxygen-17 excess ($\Delta^{17}O$) of nitrate, which is also termed mass-independent fractionation (MIF), cannot be changed or removed by subsequent mass-dependent fractionation processes and is thus conserved during atmospheric transport and processing (Brenninkmeijer et al., 2003; Vicars et al., 2013). So there will be no changes of $\Delta^{17}O$ from $HNO_3$ to $NO_3^-$. As you comment, nitrate sometimes includes both inorganic nitrate and organic nitrate in the literature. However, only inorganic nitrate is analyzed in this study. This is due to that we separated*

*dissolved inorganic nitrate from other anions (e.g., sulfate) by ion chromatography prior to analysis (He et al., 2018). According to the work of (Alexander et al., 2009), "Nitrate anion separation ensures that only inorganic nitrate is measured, assuming that soluble organic nitrate does not dissociate in water. Observations of C1-C5 alkyl nitrates in wet deposition (rain, snow, frost) (Hauff et al., 1998) suggest that they do not readily dissociate." As for whether or not organic nitrate can be used by the denitrifying bacteria (Pseudomonas aureofaciens), the work of (Hawari et al., 2000) showed that biological degradation of RDX (hexahydro-1,3,5-trinitro-1,3,5-triazine) produced $N_2O$ as a byproduct, suggesting that certain types of microorganisms can convert soluble organic nitrates into $N_2O$. However, it is not known whether or not Pseudomonas aureofaciens will do the same (Alexander et al., 2009).*

2. Correlation between $\Delta^{17}O(NO_3^-)$ and $[NO_3^-]$. It is plausible that the positive correlation is caused by that nocturnal pathways contribute more the $[NO_3^-]$. However, how to explain that the correlation is degraded when $[NO_3^-]$ is > 50 μg m$^{-3}$? Does it suggest that when $[NO_3^-]$ is high, $NO_3^-$ is not from nocturnal pathways?

*A: Thanks for your comments. We think the value of $\Delta^{17}O(NO_3^-)$ rather than the correlation between $\Delta^{17}O(NO_3^-)$ and $[NO_3^-]$ reflects the relative importance of nocturnal pathways. Take samples with $[NO_3^-]$ > 50 μg m$^{-3}$ for example, their concentration-weighted $\Delta^{17}O(NO_3^-)$ is 31.3 ‰, which corresponds to nocturnal pathways' possible fractional contribution of 56 – 100 % according to Eq. (4). This directly suggests $NO_3^-$ is mainly from nocturnal pathways when $[NO_3^-]$ is high. In fact, the correlation between $\Delta^{17}O(NO_3^-)$ and $[NO_3^-]$ mainly reflects the relationship between their variations. NOR is high (0.40±0.06) when $[NO_3^-]$ is > 50 μg m$^{-3}$, which suggests the rapid transformation of nitrate. Since visibility was always low with narrow variations (2.3±1.0 km), RH was always high with narrow range (67±7 %) and PM$_{2.5}$ was always high (201±39 μg m$^{-3}$) when $[NO_3^-]$ is > 50 μg m$^{-3}$, the relative importance of nocturnal pathways can be rather stable along the rapid transformation of nitrate, which may account for the degraded correlation.*

3. Section 3.4.4 is confusing. If coal combustion is the major contributor to NOx and coal combustion has the largest $\delta^{15}N(NO_3^-)$, why is the $\delta^{15}N(NO_3^-)$ very low (i.e., mostly ~0) in October?

*A: Thanks for your comment. The work of (Zhang et al., 2007) and (Wang et al., 2012) suggest that*

*coal combustion and vehicles are the most two important contributor to $NO_X$ annually in north China. However the relative importance of different contributors varies with time. In winter heating seasons, which lasts from mid-November to mid-March, more coal is combusted for residential heating in north China. So the relative importance of coal combustion is higher in winter heating season than that in October. Since $NO_X$ emitted from vehicles can have $\delta^{15}N(NO_X)$ smaller than 0 ‰ (Walters et al., 2015), the higher contribution from vehicles in October than in winter heating season may account for the low $\delta^{15}N(NO_3^-)$ observed in October.*

4. Many calculations are not clearly described. For example, line 214-217, it is not clear how these fractional values are calculated. Line 277, how is $[\delta^{15}N(NO_2)- \delta^{15}N(NO_X)]$ calculated? On a related note, what is the rationale to correlate $\delta^{15}N(NO_3^-)$ with $[\delta^{15}N(NO_2)- \delta^{15}N(NO_X)]$?

*A: Thanks for your comments. In the work of (Alexander et al., 2009), the fractional values are calculated by the concentration of nitrate formed through different reaction pathways divided by the total concentration of inorganic nitrate, which are all modeled by GEOS-Chem model. In the work of (Michalski et al., 2003), the fractional values are the relative proportions of $HNO_3$ production by each reaction channel, which are modeled by a zero dimensional, time dependent, photochemical box model. $[\delta^{15}N(NO_2)- \delta^{15}N(NO_X)]$ equals to the right-hand side of Eq. (6), that's $(K-1)\times(1-f_{NO2})$, where K is obtained from the work of (Walters et al., 2016) and $f_{NO2}$ is calculated by the mole concentration of $NO_2$ divided by the mole concentration of $NO_X$. Please refer to the work of (Freyer et al., 1993) for more details of the derivation process of Eq. (6). Eq. (6) suggests that $[\delta^{15}N(NO_2)- \delta^{15}N(NO_X)]$ describes the isotopic exchange between NO and $NO_2$. Since the isotopic exchange between NO and $NO_2$ can change $\delta^{15}N$ of $NO_2$, the precursor of $NO_3^-$, the positive correlation between $\delta^{15}N(NO_3^-)$ with $[\delta^{15}N(NO_2)- \delta^{15}N(NO_X)]$ is expected to suggest that the isotopic exchange between NO and $NO_2$ is likely to be an important factor for the variations of observed $\delta^{15}N(NO_3^-)$.*

Minor Comments

1. Line 118-126. Show the estimated diurnal trends in the SI.

*A: Thanks for your comment. The estimated diurnal trends are shown in Figure S1 now.*

2. Section 2.4. Discuss the purpose of using MCM estimation.

*A: Thanks for your comment. The purpose of using MCM estimation is to see whether the importance of nocturnal chemistry suggested by $\Delta^{17}O(NO_3^-)$ can be reproduced by models and to try to find potential reasons. We have added* "To see whether the relative importance of nocturnal pathways constrained by $\Delta^{17}O(NO_3^-)$ can be reproduced by models," *in line 134 before "we use the standard Master Chemical Mechanism (MCM, version 3.3, http://mcm.leeds.ac.uk/) to simulate the mixing ratios of surface $N_2O_5$ and $NO_3$ radical during our sampling period." in section 2.4*

3. Line 194-203. The authors used two methods to estimate the alpha value. These two methods should be compared and the discrepancies should be discussed.

*A: Thanks for your suggestions. We use observed $\Delta^{17}O(NO_3^-)$ to estimate the possible range of alpha, and use chemical kinetics to calculate specific alpha value to further estimate the relative importance of* nocturnal pathways. *As you know, in order to calculate specific alpha value, we estimated the concentrations of $HO_2$ and $RO_2$ radical. Our calculated specific alpha value based on the estimated concentrations of $HO_2$ and $RO_2$ radical is in the possible range of alpha constrained by observed $\Delta^{17}O(NO_3^-)$, which supports our further estimate of the relative importance of* nocturnal pathways being reliable.

4. There are many gramma errors in the manuscript. For example, line 249, add "that" after "suggest". Sentences from line 304 to 306 and from line 263-267 have many gramma errors. These two sentences are too long and should be broken down. The authors should check throughout the manuscript.

*A: Thanks for your suggestions. Grammar errors throughout the manuscript have been checked and corrected. Again, we have removed section 3.4 from the manuscript, which includes sentences from line 263-267. Sentences from line 304 to 306 have been changed into* "Calculations with the constraint of $\Delta^{17}O(NO_3^-)$ suggest that nocturnal pathways ($N_2O_5$ + $H_2O/Cl^-$ and $NO_3$ + HC) dominated nitrate production during polluted days ($PM_{2.5} \geq 75 \mu g\ m^{-3}$), with the mean possible contribution of 56 – 97 %." *in line 238-239.*

**Reference**

Alexander, B., Hastings, M. G., Allman, D. J., Dachs, J., Thornton, J. A., and Kunasek, S. A.: Quantifying atmospheric nitrate formation pathways based on a global model of the oxygen

isotopic composition (Δ17O) of atmospheric nitrate, Atmos. Chem. Phys., 9, 5043-5056, 2009.

Brenninkmeijer, C. A., Janssen, C., Kaiser, J., Röckmann, T., Rhee, T., and Assonov, S.: Isotope effects in the chemistry of atmospheric trace compounds, Chem. Rev., 103, 5125-5162, 2003.

Freyer, H. D., Kley, D., Volz‐Thomas, A., and Kobel, K.: On the interaction of isotopic exchange processes with photochemical reactions in atmospheric oxides of nitrogen, J. Geophys. Res. Atmos., 98, 14791-14796, 1993.

Hauff, K., Fischer, G., R., and Ballschmiter, K.: Determination of C1-C5 alkyl nitrates in rain, snow, white frost, lake, and tap water by a combined codistillation head-space gas chromatography technique. Determination of Henry's law constants by head-space GC, 2599-2615 pp., 1998.

Hawari, J., Halasz, A., Sheremata, T., Beaudet, S., Groom, C., Paquet, L., Rhofir, C., Ampleman, G., and Thiboutot, S.: Characterization of Metabolites during Biodegradation of Hexahydro-1,3,5-Trinitro-1,3,5-Triazine (RDX) with Municipal Anaerobic Sludge, Appl. Environ. Microbiology, 66, 2652-2657, 2000.

He, P., Alexander, B., Geng, L., Chi, X., Fan, S., Zhan, H., Kang, H., Zheng, G., Cheng, Y., Su, H., Liu, C., and Xie, Z.: Isotopic constraints on heterogeneous sulfate production in Beijing haze, Atmospheric Chemistry and Physics, 18, 5515-5528, 10.5194/acp-18-5515-2018, 2018.

Michalski, G., Scott, Z., Kabiling, M., and Thiemens, M. H.: First measurements and modeling of Δ17O in atmospheric nitrate, Geophys. Res. Lett., 30, 2003.

Morin, S., Savarino, J., Frey, M. M., Domine, F., Jacobi, H. W., Kaleschke, L., and Martins, J. M.: Comprehensive isotopic composition of atmospheric nitrate in the Atlantic Ocean boundary layer from 65 S to 79 N, J. Geophys. Res. Atmos., 114, 2009.

Vicars, W. C., Morin, S., Savarino, J., Wagner, N. L., Erbland, J., Vince, E., Martins, J. M. F., Lerner, B. M., Quinn, P. K., and Coffman, D. J.: Spatial and diurnal variability in reactive nitrogen oxide chemistry as reflected in the isotopic composition of atmospheric nitrate: Results from the CalNex 2010 field study, J. Geophys. Res. Atmos., 118, 2013.

Walters, W. W., Tharp, B. D., Fang, H., Kozak, B. J., and Michalski, G.: Nitrogen isotope composition of thermally produced NO x from various fossil-fuel combustion sources, Environ. Sci. Technol., 49, 11363-11371, 2015.

Walters, W. W., Simonini, D. S., and Michalski, G.: Nitrogen isotope exchange between NO and NO2 and its implications for δ15N variations in tropospheric NOx and atmospheric nitrate, Geophys.

Res. Lett., 43, 440-448, 2016.

Wang, S., Zhang, Q., Streets, D. G., He, K., Martin, R. V., Lamsal, L. N., Chen, D., Lei, Y., and Lu, Z.: Growth in NOx emissions from power plants in China: bottom-up estimates and satellite observations, Atmos. Chem. Phys., 12, 4429-4447, 2012.

Zhang, Q., Streets, D. G., He, K., Wang, Y., Richter, A., Burrows, J. P., Uno, I., Jang, C. J., Chen, D., Yao, Z., and Lei, Y.: NOx emission trends for China, 1995–2004: The view from the ground and the view from space, J. Geophys. Res., 112, 2007.

---

## Author Comment (AC2) · 7 Sep 2018

**Referee: J. Rudolph (Referee #2):**

**Comments:** The paper presents an interesting example for the use of isotope ratio measurements to gain insight into complex atmospheric reaction systems, here the formation of nitric acid and nitrate from NOx. Overall the paper is well written, the experimental work and interpretation solid and the subject (particle formation by oxidation of primary atmospheric pollutants is relevant for air quality. I also appreciate that the authors openly explain that isotope ratio studies in complex systems can only provide constraints (here given as range of possible contributions to nitrate formation) and that additional information is required to fully understand the magnitude of contributions from different individual reaction pathways. Consequently, I recommend publication although the authors need to address some questions and uncertainties in more detail before the paper should be accepted for publication.

*A: Thanks very much for your comments. We reply to your comments one by one as follows. One point needs to be addressed here is that we have removed section 3.4 from the manuscript due to that we are unable to explain the variations of $\delta^{15}N(NO_3^-)$ well so far.*

**Comments:** 1. May main concern is that the paper does not consider the photolysis of $NO_2$ during daytime. Although this reaction is included in Figure 1 (R3), it is not considered in the excess oxygen calculation. During daytime the reaction sequence $NO_2+h\nu=>NO+O$  $O+O_2=>O_3$  $NO+O_3=>NO_2+O_2$ (R1) will result in a steady state which can (depending on photon flux and ozone concentration) be established within several minutes. This will result not only in an isotope exchange for N between NO and $NO_2$ (Chapter 3.4.3) but also for O between $NO_X$, $O_2$ and $O_3$. In contrast to this at night R1 is a one-way street. I do not know to which extent the daytime "recycling" of NO from $NO_2$ photolysis will impact the excess oxygen ratio in $NO_2$ and NO (and consequently in nitrate) or the [15]N isotope ratio. Nevertheless, this is something that needs to be explained and discussed and potentially may change the interpretation of the isotope ratio measurements.

*A: Thanks for your comment. The work of (Michalski et al., 2014) shows that, in both the light and dark simulations of $NO_X$–$O_2$–$O_3$ system, the $\Delta^{17}O$ values between $NO_2$ and NO were essentially equal within ±0.1‰. In this case, the final $\Delta^{17}O$ value of $NO_2$ depends on the relative importance of $O_3$ oxidation in $NO_2$ production rates rather than photolysis. However, since simulation conditions have*

*difference with the ambient conditions, future work should study whether or not photolysis alone can induce large diurnal difference in $\Delta^{17}O(NO_2)$ at ambient conditions. As for the $^{15}N$ isotope ratio, previous studies suggest N isotope exchange equilibrium between NO and $NO_2$ play an important role in $\delta^{15}N$ of NO, $NO_2$ and atmospheric nitrate (Savarino et al., 2013; Freyer et al., 1993). Equation (6) suggest the partitioning of $^{15}N$ between NO and $NO_2$ depends on the relative concentration of $NO_2/NO_X$ and the temperature-dependent isotope exchange constant. During the daytime, when NO and $NO_2$ coexist in $NO_X$ cycling, the N isotope exchange between NO and $NO_2$ can influence their individual $\delta^{15}N$ (Freyer et al., 1993). At night, however, as NO is oxidized into $NO_2$ without photolysis, NO concentrations can be near zero when $O_3$ concentrations are high. In this case, $NO_2$ can reflects $\delta^{15}N$ of local $NO_X$ sources, that's $NO_2/NO_X$ approaches 1 and $[\delta^{15}N(NO_2) - \delta^{15}N(NO_X)]$ approaches 0 in Eq. (6). According to the work of (Walters et al., 2016), the lifetime of Leighton cycle reactions and $NO_X$ exchange can be comparable, therefore, the isotopic exchange between NO and $NO_2$ will be a mixture of these processes. The isotopic exchange associated with the $NO + O_3$ reaction and $NO_2$ photolysis has yet to be determined, so it will be a subject of future study. Due to that we are unable to explain the variations of $\delta^{15}N(NO_3^-)$ well, we have removed section 3.4 from the manuscript.*

**Comments:** 2.The authors use several approximations and comparisons with published results (e.g. for estimating NO, the contribution of specific pathways of nitrate formation etc.). The validity of applying these published results for this study will depend on pollution levels, degree of impact of local sources, contribution from processed polluted air masses and so on and therefore may nor be directly applicable to the cases studied here. This needs to be explained and discussed in more detail.

*A: Thanks for your suggestions. We are very sorry that some key species are not observed during our sampling period. When we use approximations to get their values, we try our best to let the approximations be reasonable or applicable for our cases. The estimate of α based on calculated $HO_2$ and $RO_2$ concentrations belongs to the first kind. Our estimated α, based on calculated $HO_2$ and $RO_2$ concentrations, is in the range of possible α values that directly derived from observed $\Delta^{17}O(NO_3^-)$ (Fig. 5) and is similar to the values determined in other mid-latitude areas (Michalski et al., 2003; Patris et al., 2007). So our estimated α on the base of calculated $HO_2$ and $RO_2$ should be reasonable. Besides, the subsequent estimate of fractional contribution of different nitrate formation pathways, which is based on estimated α and observed $\Delta^{17}O(NO_3^-)$, is a range but not a specific value. This range should*

*be representative for the real situation. We have removed section 3.4, interpretation of $\delta^{15}N(NO_3^-)$ variations, from the manuscript.*

**Comments:** 3.The various values (e.g. rate constants, excess isotope ratios in Table 2, estimates of [NO] from [CO]) used in the calculations will have uncertainties, which will add uncertainty to all quantitative results. This needs to be evaluated in more detail.

*A: Thanks for your comment. It's true that various values used in the calculations have uncertainties, and therefore add uncertainty to all quantitative results. However, as stated in the last answer, the estimated fractional contribution of different nitrate formation pathways is a range but not a specific value.*

**Comments:** 4.Subchapter 3.4.1: Indeed, the impact of deposition on $^{15}N$ is difficult to estimate. The argument that the impact of partitioning between gas and PM is minor since both $HNO_3$ and nitrate are collected on the filter is not convincing. Deposition rates for $HNO_3$ and nitrate differ and will be highly variable depending on the situation. If the $^{15}N$ isotope ratios for PM nitrate and gas phase $HNO_3$ differ, differences in deposition rates will change the isotope ratio for the sum of $HNO_3$ and nitrate.

*A: Thanks for your comment. Indeed, the impact of deposition on $^{15}N$ is difficult to estimate during long range transport. In the present study, however, our sampling site is in megacity Beijing, which is the source region for $NO_X$ and atmospheric nitrate. So the impact of deposition on our observed $\delta^{15}N(NO_3^-)$ should be minor, especially when considering that no rains were observed except for a very small snow. We agree with your comment that deposition rates for $HNO_3$ and nitrate differ. However, when considering the relatively short time of both $HNO_3$ and nitrate from being produced to being collected in our sampling site, we doubt that differences in deposition rates will not change the isotope ratio for the sum of $HNO_3$ and nitrate as much as that observed in remote areas (Geng et al., 2014). Again, we have removed section 3.4 from the manuscript.*

**Comments:** 5.Chapter 3.4.3: This chapter neglects the $NO+O_3$ and $NO_2+h\nu$ cycle (see above). Furthermore $f_{NOx}$ (in Eq. 6) is based on [NO] values calculated from measured [CO] and [$NO_2$] and consequently the calculated values for $[\delta^{15}N(NO_2) - \delta^{15}N(NO_X)]$ are in reality a non-linear function of the [$NO_2$] and [CO] concentrations. Thus Figure 7a is a plot of $\delta^{15}N(NO_3^-)$ versus a non-linear function

of [NO$_2$] and [CO]. Not sure how to interpret this, but obviously [NO$_2$] and [CO] will vary for different sources with different $^{15}$N values. In order to be of value for the reader there needs a more detailed discussion than "should therefore be interpreted with the consideration of atmospheric contexts". The discussion of $\delta^{15}$N(NO$_3^-$) should be combined into one chapter discussing the different factors that may influence $\delta^{15}$N(NO$_3^-$). Due to the complexity of the various factors influencing $\delta^{15}$N(NO$_3^-$) the attempt to discuss individual contributions separately does not work well. A revised version considering these specific problems will merit publication.

*A: Thanks very much for your comments. The influence of Leighton cycle on $^{15}$N can be summarized into the isotopic exchange constant K in Eq. (6) (Freyer et al., 1993). However, since the K value used in our study is determined from NO/NO$_2$ mixture without considering the influence of Leighton cycle (Walters et al., 2016), we truly neglects the NO+O$_3$ and NO$_2$+hv cycle. According to the work of (Walters et al., 2016), the lifetime of Leighton cycle reactions and NO$_X$ exchange can be comparable, therefore, the isotopic exchange between NO and NO$_2$ will be a mixture of these processes. The isotopic exchange associated with the NO + O$_3$ reaction and NO$_2$ photolysis has yet to be determined, so it will be a subject of future study. Due to that we are unable to explain the variations of $\delta^{15}$N(NO$_3^-$) well, we have removed section 3.4 from the manuscript.*

**Details**

General: Often a values are given as (xyz±abc), it is not always clear whether the ± indicates the error of the mean or the standard deviation.

*A: Thanks for your reminding. The ± indicates the standard deviation and it has been illustrated in the manuscript in line 17 and 141.*

Correlations: If I understand correctly, the authors present r and not r$^2$. R values of 0.5 or so correspond to r$^2$ of 0.25, a very weak correlation. These low r values need a more critical discussion of their meaning. It maybe that even a weak correlation has statistical validity. However, it has to be remembered that for r=0.5, r$^2$=0.25, which means that only 25% of the observed variability can be explained by a linear dependence between dependent and independent variable.

*A: Thanks for your comments. These low r values is not discussed for their meaning in the present manuscript.*

The authors use "wine colored" in several figure captions. Dark red would be better.

*A: Thanks for your suggestion. The "wine colored" has been changed into "dark red" throughout the manuscript.*

53: . And once formed

*A: Thanks for your suggestion. We have corrected this error in line 53.*

76: Sampling site

*A: Thanks for your suggestion. We have corrected this error in line 74.*

78: Super site set by..

*A: Thanks for your suggestion. We have corrected this mistake in line 76.*

81: About 10 km to our sampling site

*A: Thanks for your suggestion. We have corrected this mistake in line 79.*

88, 94: Insoluble substances were filtered (removed by filtration?)

*A: Removed by filter membrane.*

90: When determine the…

*A: Thanks for your suggestion. We have corrected this mistake in line 88.*

90: precision by our

*A: Thanks for your suggestion. We have corrected this mistake in line 88.*

95: which were decomposed from

*A: Thanks for your suggestion. We have corrected this mistake in line 93.*

110, 111 and other lines: is respectively

*A: Thanks for your suggestion. We have corrected this mistake.*

130: at the same time

*A: Thanks for your suggestion. We have corrected this error in line 127.*

133, 134: I assume weighted averages are meant. I understand the meaning and rational for concentration weighted oxygen excess, but I am not sure what production rate weighted means. $\alpha$ is a ratio with the total $NO_2$ production rate in the denominator, consequently the production rate weighted average for $\alpha$ would be some kind of average for the nominator, that is k[NO][O$_3$]. This requires more clarification and explanation.

*A: Thanks for your comment. The production rate weighted $\alpha$ is calculated by*

$$\frac{\sum k_{R1}[NO][O_3]}{\sum(k_{R1}[NO][O_3]+(k_{R2a}[NO][HO_2]+(k_{R2b}[NO][RO_2])}$$ *for PD of each haze event.*

164: samples

*A: Thanks for your suggestion. We have corrected this error in line 160.*

251: a small snow lasted for..

*A: Thanks for your comment. We have removed this part for the manuscript.*

258: ..it has been proposed that atmospheric nitrate that resulting from heterogeneous uptake of N…

*A: Thanks for your comment. We have removed this part for the manuscript.*

262: Don't present similar trends..

*A: Thanks for your comment. We have removed this part for the manuscript.*

518:is set by

*A: Thanks for your suggestion. We have corrected this error in line 444.*

551: . And

*A: Thanks for your suggestion. We have corrected this error in line 471.*

**Reference**

Freyer, H. D., Kley, D., Volz‑Thomas, A., and Kobel, K.: On the interaction of isotopic exchange processes with photochemical reactions in atmospheric oxides of nitrogen, J. Geophys. Res. Atmos., 98, 14791-14796, 1993.

Geng, L., Alexander, B., Cole-Dai, J., Steig, E. J., Savarino, J., Sofen, E. D., and Schauer, A. J.: Nitrogen isotopes in ice core nitrate linked to anthropogenic atmospheric acidity change, Proc. Natl. Acad. Sci. USA, 111, 5808-5812, 2014.

Michalski, G., Scott, Z., Kabiling, M., and Thiemens, M. H.: First measurements and modeling of $\Delta17O$ in atmospheric nitrate, Geophys. Res. Lett., 30, 2003.

Michalski, G., Bhattacharya, S. K., and Girsch, G.: NOx cycle and the tropospheric ozone isotope anomaly: an experimental investigation, Atmos. Chem. Phys., 14, 4935-4953, 2014.

Patris, N., Cliff, S. S., Quinn, P. K., Kasem, M., and Thiemens, M. H.: Isotopic analysis of aerosol sulfate and nitrate during ITCT‑2k2: Determination of different formation pathways as a function of particle size, J. Geophys. Res. Atmos., 112, 2007.

Savarino, J., Morin, S., Erbland, J., Grannec, F., Patey, M. D., Vicars, W., Alexander, B., and Achterberg, E. P.: Isotopic composition of atmospheric nitrate in a tropical marine boundary layer, Proc. Natl. Acad. Sci. USA, 110, 17668-17673, 2013.

Walters, W. W., Simonini, D. S., and Michalski, G.: Nitrogen isotope exchange between NO and NO2 and its implications for $\delta15N$ variations in tropospheric NOx and atmospheric nitrate, Geophys. Res. Lett., 43, 440-448, 2016.

---

## Author Comment (AC3) · 7 Sep 2018

**G. Michalski (Referee #3):**

**Comments:** A very interesting and exciting dataset. I think the manuscript would do well with some significant revisions.

*A: Thanks very much for your comments. We reply to your comments one by one in the following part. One point needs to be addressed here is that we have removed section 3.4 from the manuscript due to that we are unable to explain the variations of $\delta^{15}N(NO_3^-)$ well so far.*

**Comments:** Line 114 it is unclear to what the coefficients 24.85 and 13.66 mean or where they are derived. As someone versed in the field, and some information on line 26, I can surmise this is the $\Delta^{17}O$ value $NO_2+OH$ pathway, but this is in no way clear to the non-specialist. There are host of assumptions that go into this number that are not explained and have uncertainties that are not being propagated through. Six points on this are

1. From the text there is the assumption that the $\Delta^{17}O$ of $O_3$ is essentially a fixed value of 26‰, which is by no means codified in the literature, despite the some who would hope so because it makes the data analysis less problematic.

2. Johnston et al. and Krankowsky et al. observed $O_3$ $\Delta^{17}O$ values that spanned 18.8‰ to 41‰ with a standard deviation of 4.8‰.

3. Two papers by the Savarino group using a different method arrive at values close to 26‰ with smaller variations of 1 and 1.6‰. Their Antarctic paper noted $O_3$ $\Delta^{17}O$ had "insignificant variation" 28 ‰ - 23 ‰, if one considers ~20% variation insignificant.

4. Lab experiments have clearly noted an $O_3$ $\Delta^{17}O$ temperature dependence.

5. $NO_X$ photochemical equilibrium experiments (Michalski et al., 2013) a higher terminal atoms value transfer and Vicars noted that $\Delta^{17}O(O_3)$trans. in the range of 38–44‰ fits data.

6. Even assuming a fixed value of $O_3$ $\Delta^{17}O$ value of 26‰, one cannot increase significant (24.85) digits by division/multiplication.

The authors should note these conflicting assumptions and how these assumptions would influence their interpretations of reaction pathways.

*A: Thanks for your comment. The value of 24.85α and 24.85α + 13.66 in line 114 is respectively the $\Delta^{17}O$ value $NO_2+OH$ and NO3+HC pathway (Table 1). To be clear for readers, we have added "By*

*using the $\Delta^{17}O$ assumptions for different pathways in Table 1 and the definition $f_{R6} + f_{R7} + f_{R8} + f_{R9} + f_{R10} = 1$, Eq. (1) is further expressed as:"* in line 110 before Eq. (2). And to be consistent with the significant digit of our assumption ($\Delta^{17}O(O_3) = 26$ ‰), "24.85" and "13.66" have been changed into "25" and "14" respectively throughout the manuscript. We have learned that observed $\Delta^{17}O$ values spanned largely in the work of (Krankowsky et al., 1995) and (Johnston and Thiemens, 1997) during the preparation of our manuscript. However, (Vicars and Savarino, 2014) questioned in their paper that *"In the study of Krankowsky et al. (1995), no correlation was found between the $\delta^{17}O$ and $\delta^{18}O$ values of ozone, suggesting that the large degree of variability observed for $\Delta^{17}O$ is an artifact resulting from statistical scatter of the individual d measurements. These results are therefore not inconsistent with the hypothesis that the tropospheric value of $\Delta^{17}O(O_3)$ is constant. However, the data of Johnston and Thiemens (1997) reveal a systematic variation in the relationship between $\delta^{17}O$ and $\delta^{18}O$, with data from three different sites aligning on different slopes in a three-isotope plot. The authors of this study concluded that the observed variations resulted from differences in ozone transformation pathways between the three sites and suggested that measurements of the triple-isotope composition of ozone could therefore be useful in constraining the tropospheric ozone budget. This conclusion was later questioned by Brenninkmeijer et al. (2003), who argued that the differences in slope were not statistically significant and suggested that they were related to analytical bias."* In addition, $\Delta^{17}O(O_3) \approx 26$ ‰ from the observations of (Vicars and Savarino, 2014) and (Ishino et al., 2017) compare quite well in terms of average value: $25 \pm 11$ ‰ and $26 \pm 5$ ‰ for the studies of Krankowsky et al. (1995) and Johnston and Thiemens (1997) respectively, and the observations of (Vicars and Savarino, 2014) and (Ishino et al., 2017) are more recent publications, so we prefer $\Delta^{17}O(O_3)$ values reported by (Vicars and Savarino, 2014) and (Ishino et al., 2017). The assumption that $\Delta^{17}O(O_3) \approx 26$ ‰ is also adopted by (Chen et al., 2016). It's true that lab experiments have clearly noted an $O_3$ $\Delta^{17}O$ temperature dependence. However, as (Vicars and Savarino, 2014) summed in their paper, *"the experimentally determined dependency of $\Delta^{17}O(O_3)$ on the pressure of ozone formation suggests a relatively small decrease of only ~2 ‰ for an increase in pressure from 500 to 760 Torr (0.7 to 1.0 atm) (Morton et al.,1990; Thiemens and Jackson, 1990); and temperature dependency studies suggest an increase in $\Delta^{17}O$ of only ~5 ‰ for an increase in ozone formation temperature from 260 to 320 K (Morton et al., 1990; Janssen et al., 2003). For these reasons, it is often assumed that $\Delta^{17}O(O_3)_{bulk}$ in the troposphere exhibits no more than a 1–2 ‰ level of variability under standard*

*surface conditions". Nevertheless, we noted that both (Vicars and Savarino, 2014) and (Ishino et al., 2017) uses the nitrite-coated filter technique in their studies, future studies may need other technique to verify whether $\Delta^{17}O(O_3)$ is truly constant in the surface atmosphere.*

**Comments:** NO was derived CO mixing ratios derived from observations in winter Beijing (Lin et al., 2011). The correlation coefficients for this relationship are .76 and .82, which means there is some uncertainty in the derived NO. How would this impact the author's results

*A: Thanks for your comment. We realized that we are unable to explain $\delta^{15}N(NO_3^-)$ data well so far, and thus removed section 3.4 from the manuscript.*

**Comments:** Line 200: "To estimate the specific α value, chemical kinetics in Table 2 and Eq. (3) were used. Specific α is estimated to range from 0.86 to 0.97 with a mean of (0.94±0.03)". The coefficients used to estimate $HO_2$ has significant uncertainties (again r2= ~0.7) and the regression itself is are not universal but are valid for Tokyo. No discussion on whether this would hold in an extreme haze event in Beijing. Likewise the uncertainty of $RO_2 = 0.7HO_2$ must be significant and site specific. The validity of this assumption in the context of extreme haze needs to be discussed.

*A: Thanks for your comment. As we all know, there are some similarities between Tokyo and Beijing, e.g., both of them are in the East and both of them are megacities, which increases the possible applicability of using the regression. In the regression, the $HO_2$ concentration is related with $O_3$ concentration (Kanaya et al., 2007), and we expect $HO_2$ concentration should be related with $O_3$ concentration too in Beijing as both $HO_2$ and $O_3$ are photochemical products whether or not in haze. Meanwhile, in the same season, the $HO_2$ concentration observed in Beijing (Liu et al., 2012) is generally comparable with that reported by (Kanaya et al., 2007) in Tokyo. If we double the estimated $HO_2$ and $RO_2$ concentrations, the calculated α would be 0.89±0.05. If we halve the estimated $HO_2$ and $RO_2$, the calculated α would be 0.97±0.02. Both of these two situation will not change the importance of nocturnal chemistry reported in the manuscript. As for $RO_2 = 0.7HO_2$, it's the general value reported in the literature (Liu et al., 2012; Elshorbany et al., 2012; Mihelcic et al., 2003). Neither double nor halve the value will change the importance of nocturnal chemistry reported in the manuscript (α = 0.92±0.04 and 0.95±0.02 respectively). In addition, Our estimated α, based on calculated $HO_2$ and $RO_2$ concentrations, is in the range of possible α values that directly derived from*

*observed $\Delta^{17}O(NO_3^-)$ (Fig. 5) and is similar to the values determined in other mid-latitude areas (Michalski et al., 2003; Patris et al., 2007). So our estimated α on the base of calculated $HO_2$ and $RO_2$ should be reasonable.*

**Comments:** "lifetime of atmospheric nitrate is typically on the order of days (Vicars et al., 2013)" I doubt that Vicars was the first to determine the lifetime of nitrate in the atmosphere. Further the lifetime is significantly dependent on precipitation frequency so if there was no rain during the collection period the lifetime of nitrate is significantly longer, though it does not change the authors point.

*A: Thanks for your reminding. We have changed the reference into an earlier one, i.e., (Liang et al., 1998).*

**Comments:** 138 "We use the Master Chemical Mechanism "This requires an entire discussion section. MCM is a gas phase mechanism. Were heterogeneous reactions included? Based on what uptake scheme? How aerosol surface area was determined if that was part of the scheme? "1-h averaged mixing ratios of observed surface CO, $NO_2$, $SO_2$ and $O_3$ and estimated NO" what does this mean? Did you initialize the model with these mixing ratios? Or did you correct the model to match these hourly? Or did you run the model hourly? What length of spin-up do you use? How was photolysis adjusted to account for haze? This model predicts things like $OH_2$, $RO_2$, NO…how does the model prediction compare with your estimation of these key compounds that we parameterized by your isotope scheme, but not measured? How does it predict things that were measured over time (O3, $NO_2$,)? This section was entirely too vague for anything useful to be inferred about the accuracy of predicted NO3 or N2O5 mixing ratios.

*A: Thanks very much for your comments. The MCM model (version 3.3) we used is the standard one from the website (http://mcm.leeds.ac.uk/). The model includes heterogeneous reactions. However, we have no aerosol surface data as input. The 1-h averaged mixing ratios of observed surface CO, $NO_2$, $SO_2$ and $O_3$ and estimated NO is used to initialize the model and these mixing ratios are updated every 12 hours. The model is set to output one dataset per hour. We did not adjust the photolysis to account for haze, so the model predicted $HO_2$ and $RO_2$ is expected to be higher than the real value. In fact, the average of model predicted $HO_2$ during the sampling period (including day and night) is 1.35 ppt,*

*higher than our estimated value (0.88 ppt) by ~50%. Therefore we used the estimated value rather than the model predicted HO₂ in our calculation due to that the estimated value is based on observed O₃ concentration. There also exist gaps between the measured O₃, NO₂ and predicted O₃, NO₂ (17 and 31 ppb vs 26 and 23 ppb respectively). This may due to that photolysis was not adjusted and the emission of NOₓ was not considered during modeling. Since we use the standard MCM model only to get nocturnal radicals (N₂O₅ and NO₃), the unadjusted photolysis may be not a major factor influencing predicted NO₃ or N₂O₅ mixing ratios. In addition, the variation trend of predicted NO₃ and N₂O₅ is a more useful information than the specific concentration in our study, which possibly deduce the risk of using this model in the present study.*

**Comments:** "variation of atmospheric $\delta^{15}N(NO_3^-)$ can be interpreted by the following four processes (Vicars et al.,2013)" again please give credit where credit is due, Freyer used this scheme 20 years before Vicars to investigate 15N variations in atmospheric nitrate.

*A: Thanks for your reminding. This reference has been replaced by (Freyer, 1991).*

**Comments:** 254 "The quartz filter used here is thought to collect both particulate nitrate and gaseous HNO3" this statement needs better justification by citing filter pack studies. This is particularly true in Beijing where NH4NO3 is a major component of PM and loss by volatilization could also be occurring. Vicars, like myself (2003), limited this assumption to coastal sampling where seas salt buffering was present and noted that "the exact nature of the nitrate species collected during sampling using glass fiber filters has always been an area of some debate due primarily NH4NO3.

*A: Thanks very much for your reminding. We realized that the exact nature of the nitrate species collected during sampling using fiber filters has always been an area of some debate due primarily NH₄NO₃ and thus removed the statement from the manuscript.*

**Comments:** Isotopic fractionations associated with nitrate formation pathways. These (Photolysis and KIE effects in NOy) are largely unknown and the discussion should reflect that. Walters ab initio paper indicates IF equilibrium is dominant the more oxidized compounds should have higher $^{15}$N. Is this consistent with observations?

*A: Thanks for your comment. It's true that isotopic fractionations associated with nitrate formation*

*pathways are largely unknown, so we decided to remove the entire section 3.4 from the manuscript. As for our observation, $\delta^{15}N(NO_3^-)$ is generally high (7.4±6.8 ‰), however, we do not know whether it is related to nitrate formation pathways. The $\delta^{15}N(NO_3^-)$ data is open for you if you are interested in haze in China.*

**Comments:** 275 "Where K is the isotopic exchange constant of N between NO and $NO_2$, which is temperature-dependent .." It is not clear if the authors are using temperature to calculate this daily, if so what temperature? Average? Day and night average? Clearly this equation is very dependent on fraction of $NO_2$, which is based on NO estimations that also have uncertainty, which should be discussed and represented on the y-axis error bar on figure 7. That caption should emphasize the Y data is not a measurement of the $\delta15N$ of ambient $NO_X$ (Freyer, Walters) but a calculation. It would also seem that since the authors are presenting $\delta^{15}N$ in ‰, that the RHS of Eq, 6 will need to be multiplied by a factor of 1000.

*A: Thanks for your comment. We uses the 12h-averaged temperature to calculate this. We cannot know how much the uncertainty of NO estimation influences the relationship between $\delta^{15}N(NO_3^-)$ and $[\delta^{15}N(NO_2) – \delta^{15}N(NO_X)]$, so we removed the entire section 3.4 from the present manuscript.*

**Comments:** 279 "the correlation is better in residential heating season … especially in residential heating season. " mechanistic, why would this so? The authors seem to imply residential heating is promoting exchange when its likely NO/NO2 ratios. Was the a correlation between $\delta^{15}N$ and $f_{NO2}$? The exchange section should discuss in terms of Freyers and Walters et al. papers that measured $\delta^{15}N$ values of ambient $NO_2$.

*A: Thanks for your comment. I have no idea why the correlation is better in residential heating season, perhaps due to that source emission in residential heating season is more stable, leading to other factors, e.g., isotopic exchange, being more important for the trend of $\delta^{15}N(NO_3^-)$. Again, we removed the entire section 3.4 from the present manuscript.*

**Comments:** "Influence of $NO_X$ emissions." This section could be greatly expanded, there has been a lot of recent work by the Elliot, Hastings, and Michalski groups of $^{15}N$ sources. While coal maybe be dominant in the surrounding regions, automobiles and diesel trucks in Beijing must be significant,

particularly during stagnant conditions. Is there a better N inventory for Beijing itself ?

*A: Thanks very much for your suggestions. It's true that coal combustion and vehicles are the most important emissions in Beijing and its surrounding regions. We are sorry that we have not found better N inventory for Beijing, perhaps Qiang Zhang in Tsinghua University have the last inventory for Beijing.*

**Comments:** I did not see any discussion about any (or lack thereof) correlation between $\delta^{18}O$ $\Delta^{17}O$ and $\delta^{15}N$. If they are completely decoupled then that would argue for source effects, if there is some covariation, then exchange/chemistry could be the main process.

*A: Thanks for your comments. There is no correlation between $\Delta^{17}O$ and $\delta^{15}N$ (Fig. 4f), so we did not further discuss their relationship. The $\delta^{18}O$ is highly positively correlated with $\Delta^{17}O$ ($R^2 = 0.9$, data not shown), which means it may have almost the same implications with $\Delta^{17}O$, and thus we did not present the data of $\delta^{18}O$ but $\Delta^{17}O$ in the manuscript.*

**Reference**

Chen, Q., Geng, L., Schmidt, J. A., Xie, Z., Kang, H., Dachs, J., Cole-Dai, J., Schauer, A. J., Camp, M. G., and Alexander, B.: Isotopic constraints on the role of hypohalous acids in sulfate aerosol formation in the remote marine boundary layer, Atmos. Chem. Phys., 16, 11433-11450, 2016.

Elshorbany, Y. F., Kleffmann, J., Hofzumahaus, A., Kurtenbach, R., Wiesen, P., Brauers, T., Bohn, B., Dorn, H. P., Fuchs, H., and Holland, F.: HOx budgets during HOxComp: A case study of HOx chemistry under NOx‐limited conditions, J. Geophys. Res., 117, 2012.

Freyer, H. D.: Seasonal variation of 15N/14N ratios in atmospheric nitrate species, Tellus B, 43, 30-44, 1991.

Ishino, S., Hattori, S., Savarino, J., Jourdain, B., Preunkert, S., Legrand, M., Caillon, N., Barbero, A., Kuribayashi, K., and Yoshida, N.: Seasonal variations of triple oxygen isotopic compositions of atmospheric sulfate, nitrate, and ozone at Dumont d'Urville, coastal Antarctica, Atmos. Chem. Phys., 17, 3713-3727, 2017.

Johnston, J. C., and Thiemens, M. H.: The isotopic composition of tropospheric ozone in three environments, J. Geophys. Res. Atmos., 102, 25395-25404, 1997.

Kanaya, Y., Cao, R., Akimoto, H., Fukuda, M., Komazaki, Y., Yokouchi, Y., Koike, M., Tanimoto, H.,

Takegawa, N., and Kondo, Y.: Urban photochemistry in central Tokyo: 1. Observed and modeled OH and HO2 radical concentrations during the winter and summer of 2004, J. Geophys. Res., 112, 2007.

Krankowsky, D., Bartecki, F., Klees, G., Mauersberger, K., Schellenbach, K., and Stehr, J.: Measurement of heavy isotope enrichment in tropospheric ozone, Geophys. Res. Lett., 22, 1713-1716, 1995.

Liang, J., Horowitz, L. W., Jacob, D. J., Wang, Y., Fiore, A. M., Logan, J. A., Gardner, G. M., and Munger, J. W.: Seasonal budgets of reactive nitrogen species and ozone over the United States, and export fluxes to the global atmosphere, J. Geophys. Res. Atmos., 103, 13435-13450, 1998.

Liu, Z., Wang, Y., Gu, D., Zhao, C., Huey, L., Stickel, R., Liao, J., Shao, M., Zhu, T., and Zeng, L.: Summertime photochemistry during CAREBeijing-2007: ROx budgets and O3 formation, Atmos. Chem. Phys., 12, 7737-7752, 2012.

Michalski, G., Scott, Z., Kabiling, M., and Thiemens, M. H.: First measurements and modeling of Δ17O in atmospheric nitrate, Geophys. Res. Lett., 30, 2003.

Mihelcic, D., Holland, F., Hofzumahaus, A., Hoppe, L., Konrad, S., Müsgen, P., Pätz, H. W., Schäfer, H. J., Schmitz, T., and Volz‐Thomas, A.: Peroxy radicals during BERLIOZ at Pabstthum: Measurements, radical budgets and ozone production, J. Geophys. Res., 108, 2003.

Patris, N., Cliff, S. S., Quinn, P. K., Kasem, M., and Thiemens, M. H.: Isotopic analysis of aerosol sulfate and nitrate during ITCT‐2k2: Determination of different formation pathways as a function of particle size, J. Geophys. Res. Atmos., 112, 2007.

Vicars, W. C., and Savarino, J.: Quantitative constraints on the 17O-excess (Δ17O) signature of surface ozone: Ambient measurements from 50°N to 50°S using the nitrite-coated filter technique, Geochim. Cosmochim. Acta, 135, 270-287, 2014.